# ENHANCEMENT OF GNN'S EXPRESSIVE POWER VIA RECONSIDERING MODAL LOGIC

## ABSTRACT

Since AC-GNNs, in which nodes only gather information from their neighbors to update features at every layer, are limited in their expressive power, numerous models have been proposed to enable GNNs to go beyond Weisfeiler-Lehman (WL) test. However there still a lack of effective methods to measure these models' expressive power: for a specific task, it is still difficult to figure out whether the model is competent for the task. We tackle this problem by finding equivalent Boolean classifiers logic for models. By checking whether the task is able to be expressed as model's equivalent Boolean classifiers logic formula, we can be aware of whether the model is competent for task. We propose a framework for AC-GNNs, denoted as l-div AC-GNNs, to enhance the expressive power. To be more specific, we classify node's neighbors according to existence of different length of paths from node's neighbors to itself. To find l-div AC-GNNs' equivalent Boolean classifiers logic, we introduce the l-div graded modal logic and prove that a Boolean logical classifiers can be expressed by l-div graded modal logic if and only if there exists a l-div AC-GNN which is able to capture it. In this paper, three properties are defined for a framework: invariance and equivariance, approximation and logic expressive power, we proved l-div AC-GNNs are possessing with these properties. A series of tasks have been implemented to validate our theoretics, the results of experiments demonstrate the validities of both our method to measure models' expressive power and expressive power of l-div AC-GNNs.

## 1 INTRODUCTION

Graph neural networks (GNNs) (Merkwirth & Lengauer, 2005)(Hamilton et al., 2017)(Velickovic et al., 2017) have become recently become popular and powerful methods with a wide range of applications, for graph representation learnig, as social and financial networks(Yang et al., 2021), molecule classification for chemomatics(Dreier et al., 2000), knowledge graph(Yasunaga et al., 2021) analysis and Web page ranking (Qi & Davison, 2009). Most GNNs follow neighborhood aggregate-conbine (or message passing) strategy, as at each iteration node will gather feature information of its neighbors' to update its new feature representation (Scarselli et al., 2008). Since the expressive power of aggregate-conbine GNNs(we denote as AC-GNN) is bounded by bounded by the one-dimensional Weisfeiler-Lehman (1-WL) test (Xu et al., 2018), there still substantial tasks that Message-passig GNNs will fail to accomplish. In this paper, we propose a new framework for AC-GNN (denoted as l-div AC-GNN) to enhance the expressive power of represent AC-GNN, through establishing connection with model theory formulas to understand its expressive power.

Xu (Xu et al., 2018) has revealed the the theoretical connection between expressive power of GNN and the Weisfeiler-Lehman (WL)(Douglas, 2011) test algorithm. At each iteration WL algorithm gather color of nodes' neighbors' color to update node color, and it will decide whether a pair of graph is isomorphism by checking whether there is bijection between the label of each node of two graphs. We found out that WL test algorithm sometimes decides some pairs of non-isomorphism graphs is isomorphism because it lacks the ability to recognize different type of node's neighbors. Based on this, we have proposed a new graphs isomorphism test algorithm we call it l-div color refinement algorithm, it classifies node's neighbors according to whether the neighbor is node's $i^t h$ hop neighbor. The ability to discriminate different types of neighbor enable l-div color refinement provably to distinguish more non-isomorphism graphs than Weisfeiler-Lehman (WL) test algorithm.

Recently, several directions has been proposed to enhance GNN's expressive power:(1) adding random feature into model (Vignac et al., 2020)(Sato et al., 2019) (2) transform GNNs into higher-order (i.e. k-WL with k 3) (Maron et al., 2018)(Maron et al., 2019) (3) adding local pre-defined graph substructures information as additional features (Bouritsas et al., 2022)( A New 2022); Different from these work, based on l-div color refinement algorithm, we propose a general framework solution for every AC-GNN to enhance its expressive power to capture neighborhood structural properties of graphs. For most of these models, they sacrifice complexity of computation or accuracy (due to adding randomness into model) in exchange for higher expressive power. However, our framework overcomes this limitation compared with methods in (1), adding randomness will effect on the accuracy, our framework is able to stay invariant for same input features while doesn't require high computational complexity and still go beyond WL-test. Compared with methods in (2), k-GNN and k-foreloard GNN for example, with improvement of expressive power, the space and time complexity will also grow exponentially, as requiring $O(n^{k+1})$ time complexity and $O(n^k)$ space complexity. Compared with methods in (3), most methods still require high space complexity and time complexity and may only be fit for limited application, our framework is suitable for every AC-GNN and preserve time complexity and space complexity for $O(2^{l-1}n^k)$.

Pablo (Barceló et al., 2020) show the connection between First logic and AC-GNNs and has characterized exactly every formula that can be captured by AC-GNNs if and only if it expressed as graded logic. In order to enhance formula expressive power, We propose l-div graded logic while we have proved every graded logic can be expressed by l-div graded logic. It refines graded logic binary relation $E(x, y)$ into $E_{(i_2, \cdots i_l)}(x, y)$. Also, we have establish the relationship between l-div graded logic and l-div AC-GNN, by proving that every formula that can be expressed by l-div AC-GNNs if and only if it can be expressed as l-div graded logic. Therefore, to evaluate whether the l-div framework is competent for a task, we just have to check whether the task is able to expressed by l-div graded logic.

The following are our main contributions:

(1)We have proposed a new algorithm l-div color refinement for graphs isomorphism test, and proved it is more powerful than WL-test.

(2)We have proposed a new framework as l-div AC-GNNs, and proved it is at most as powerful as l-div color refinement algorithm, hence the framework is more expressive than AC-GNNs. We proposed three properties that improved framework should qualified, and proved l-div AC-GNNs has the properties.

(3)We have proposed the l-div graded logic, and proved every formula that can be expressed by l-div AC-GNNs if and only if it can be expressed as l-div graded logic, and propose a procedure for evaluating whether the model is fit for the task.

(4) We proposed a new framework for model to measure their expressiveness power by finding their equivalent modal logic. We have found the equivalent modal logic for k-GNN,local k-GNN, GSN ESAN and SPD-WL.

We experimentally validate our theoretical result by showing the result of task for AC-GNNs and l-div AC-GNNs. In particular, we use synthetic graph data to detect whether the node is contained by triangle which can be expressed by 2-div graded logic while not by graded logic. As the result, the three common AC-GNNs: GCN, GIN, TAG struggle to fit the training data while 2-div AC-GNNs is able to generalize with 100 % accuracy. For counting number of triangles that node is contained, even though the task can not be expressed by 2-div graded model logic, the performances of 2-div AC-GNNs are still better than AC-GNNs.

## 2   A NEW HIERARCHY OF NODE'S NEIGHBOR CLASSIFICATION

In this section, we will introduce the hierarchy of node's in-neighbor classification based on node's local structure of the graph. Let graph $\vec{G} = (V, \vec{E})$ be a directed graph with node set $V$ and directed edges set $\vec{E}$, $n = |V|$ represents the number of nodes. We only consider simple directed graph and node's in-neighbors, since every simple undirected graph G=(V,E) can be regarded as directed graph with node set V and edge set $\vec{E} = \{\overrightarrow{(v, u)}, \overrightarrow{(u, v)} | (u, v) \in E\}$, and simplify in-neighbor as neighbor.

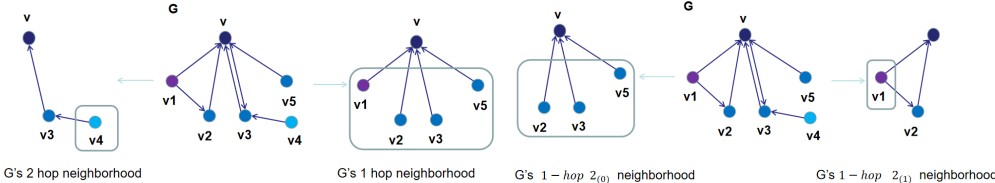

Figure 1: An overview of our classification for nodes' neighbors. v1,v2,v3,v5 are v's 1-hop neighbors and v4 is v's 2-hop neighbor. Also v1,v2,v3,v5 are classified into two different 1-hop neighbors types as 1hop $2_{(0)}$ in-neighborhood: v2,v3,v5 and 1hop $2_{(1)}$ in-neighborhood: v1, which will provide more information for model to capture node's local structure.

The set of $\mathcal{N}(v) = \{u \in V | \overrightarrow{(v, u)} \in \vec{E}\}$ denotes in-neighbors of a vertex v. Denote $A \in N^{n \times n}$ as the adjacency matrix of graph $\vec{G}$ where $a_{u,v} \in A$ represents there are $a_{u,v}$ edge $\overrightarrow{(u, v)}$ from node u to node v. In the following, we define notions of the sets of node's neighborhoods and the adjacency matrix induced by them, which are connected with node's local structure. If not specially mentioned, we denote direct graph as $G = (V, E)$ instead of $\vec{G} = (V, \vec{E})$ and regard undirected graph as a directed graph with edge set $\vec{E} = \{\overrightarrow{(v, u)}\}$.

**Definition 2.1** (K-Hop Neighborhood). *Node u is said to be node v's $k - hop$ neighborhood, if the length of the shortest directed path from u to v equals to K. We denote the set of u as $N^{k-hop}(v)$ or $N(v)$ when k=1.*

**Definition 2.2** (K-Hop $\boldsymbol{L_{(i_{(K+1)} \cdots i_L)}}$ Neighborhood). *Node u is said to be node v's $k - hop$ $L_{(i_{(k+1)} \cdots i_L)}$ neighbor $(i_{(k+1)} \cdots i_l \in (0, 1)^{L-K})$, then for any t $(K + 1 \leq t \leq L)$, $i_t = 1$ if and only if then there exists directed path with length t from u to v, else if $i_t = 0$ then there does not exist any directed path with length t from u to v. Denote the set of u as $N^{k-hop}_{L_{(i_1, i_2 \cdots i_t)}}(v)$ or $N_{L_{(i_1, i_2 \cdots i_t)}}(v)$ when k=1.*

**Definition 2.3** (K-Hop $\boldsymbol{L_{(i_{(K+1)}, i_{(K+2)} \cdots i_L)}}$ Induced Adjacency Matrix). *Given a graph with adjacency matrix A, Call $A^k$ the k-hop adjacency matrix and denote $k - hop$ $l_{(i_{(k+1)}, i_{(k+2)} \cdots i_L)}$ adjacency matrix as $A^k_{l_{(i_{(k+1)}, i_{(k+2)} \cdots i_l)}}$ where $(A^k_{l_{(i_{(k+1)} \cdots i_l)}})_{u,v} = 0$ if and only if $u \notin N^{k-hop}_{L_{(i_{(k+1)} \cdots i_l)}}(v)$, else $(A^k_{L_{(i_{(k+1)} \cdots i_l)}})_{u,v} = 1$ if $u \in N^{k-hop}_{L_{(i_{(k+1)} \cdots i_l)}}(v)$.*

**Definition 2.4.** *We define operator $Mask_A(\cdot)$ and $Mask_{\sim A}(\cdot)$ as follow:*

$$Mask_A(X)_{i,j} = \begin{cases} 0 & if \quad a_{i,j} = 0 \\ x_{i,j} & if \quad a_{i,j} \neq 0 \end{cases} \tag{1}$$

$$Mask_{\sim A}(X)_{i,j} = \begin{cases} x_{i,j} & if \quad a_{i,j} = 0 \\ 0 & if \quad a_{i,j} \neq 0 \end{cases} \tag{2}$$

Also, we define the direct product of operator $Mask_A(\cdot)$ as: $Mask_{A \odot B}(\cdot) = Mask_A(Mask_B(\cdot))$ and $Mask_{\prod_{i=1}^k \odot A_i}(\cdot) = Mask_{A_k \odot A_{k-1} \odot \cdots \odot A_1}(\cdot)$

**Lemma 2.1.** *Let $\vec{G}$ be a graph with adjacency matrix A. The number of walks from u to v in $\vec{G}$ with length k is $(A^k)_{u,v}$*

**Theorem 2.1.** *Given sequence $(i_{(k+1)}, i_{(k+2)} \cdots, i_l) \in (0, 1)^{(l-k)}$, a graph $G = (V, E)$ with induced adjacency matrix A and t$(k + 1 \leq t \leq l)$, define $(A^t)^{(i_{(k+1)} \cdots, i_l)} = A^t$ if $i_t = 1$, else $(A^t)^{(i_{(k+1)} \cdots, i_l)} = \sim A^t$ if $i_t = 0$, then $A^K_{L_{(i_{(k+1)} \cdots, i_l)}} = Mask_{\prod_{t=K+1}^L \odot (A^t)^{(i_{(k+1)} \cdots, i_l)}}(A^K)$*

The proofs of lemma 1 and theorem 1 are provided in Appendix C.

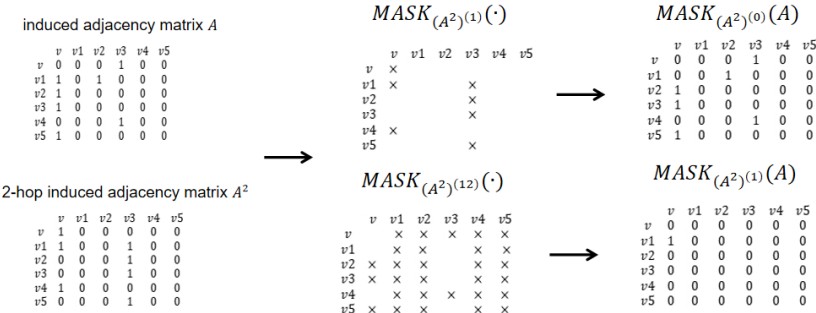

Figure 2: Figure 2 shows how to compute $1 - hop\ 2_{(0)}$ and $1 - hop\ 2_{(1)}$ induced adjacency matrix $A_{2_{(0)}}, A_{2_{(1)}}$ : First, compute 2-hop induced adjacency matrix $A^2$ to generate operator $MASK_{(A^2)^{(0)}}(\cdot)$ and $MASK_{(A^2)^{(1)}}(\cdot)$,then by theorem 1. we are able to gain $1 - hop\ 2_{(0)}$ and $1 - hop\ 2_{(1)}$ induced adjacency.

# 3 L-DIV GRADED MODEL LOGIC

## 3.1 INTRODUCTION OF GRADED MODEL LOGIC

Over a graph, first order (FO) logic classifier regards every node as a "term". If every node has a label as $label(x)$, consider a formula classifier $\alpha(x)$ that is formed with atomic formulas: $x = y$, $E(x,y)$, $label(x)$, boolean connectives: $\wedge, \neg$ as "and" and "not" and quantifiers $\exists$ as Boolean classifiers. Also, first order (FO) logic is able to represent the boolean connectives $\vee, \rightarrow, \leftrightarrow$ as "or","implies" and "iff" respectively. Consider the formula with one $free\ variable$ as follow:

$$\beta(x) := \text{Yellow}(x) \wedge \exists y\big(E(x,y)\big) \wedge \exists z\big(E(x,z) \wedge \text{Red}(z)\big) \wedge \neg(y = z) \tag{3}$$

The logic classifier is $True$ iff node x's label is yellow and has at least two neighbors while at least one of them whose label is red. The extension FOC of first order logic replaces the quantifiers by counting quantifiers: $\exists^{\geq p}, \forall^{\geq p}$ like the following formula:

$$\gamma(x) := \text{Yellow}(x) \wedge \exists^{\geq 4}y[\,E(y,x) \wedge \text{Green}(y)] \tag{4}$$

Since $FOC$ is too powerful to make connection with GNN, $FOC_2$ is introduced to restrict the expressive power by allowing at most two of the variables are used in the formula, the set of formula is denoted as $FOC_2$

Since any node for a fixed number L of layers in an AC-GNN is not able to learn any information further than at distance L in the graph, hence there exists formula $\alpha \in FOC_2$ that is not able to be captured by any GNN with a fixed number L of layers.

**Proposition 1.** *(Barceló et al., 2020) There is an FOC2 classifier that is not captured by any AC-GNN*

Therefore, pablo introduced graded modal formulas to restrict formula expressive power. Graded modal formulas are formed with propositional variables $p, q \cdots$, boolean connectives $\wedge, \neg$ as "and" and "not", and the unary modal operators $\Diamond^{\geq n}$ for $n > 1$, the set Graded modal formulas is denoted as $L_{GML}$.

A model(Helfand, 1975) for graph is a triple $M = (V, R, W)$, where V is a non-empty set of nodes, R is a binary relation on V, in graph $xRy$ usually denotes there is an directed edge from node y to x, and W is a valuation, which is a function mapping a subset of V to every proposition letter. For example, if there is a path $(v_1, v_2, \cdots v_n)$, we can define $W(v_1, v_2, \cdots v_n)$ to denote the length of the path. We can see graded modal formulas is formed in a familiar way for the atomic formula $W(v_1, v_2, \cdots v_n) = q$ and boolean connectives $\neg\ \wedge$ , and the modal operators is defined as follow:

$$M, v \models \Diamond^n \varphi \iff \exists^{\neq} v_1 \ldots v_n \bigwedge_{1 \leq i \leq n} (vRv_i \wedge M, v_i \models \varphi) \tag{5}$$

In direct perceiving, graded modal formula only allows formula to check the property of node's neighbor or itself, which is similar to the framework of GNNs. Hence if there is not a path from x to y, then formula $\alpha$ is not able to be expressed by graded modal formulas.

**Definition 3.1.** *A GNN classifier A captures a logical classifier $\varphi(x)$ if for every graph G and node v in G, it holds that $A(G, v) = true$ if and only if $(G, v) \models \varphi$.*

**Proposition 2.** *((Barceló et al., 2020)) A logical classifier is captured by AC-GNNs if and only if it can be expressed in graded modal logic.*

Figure 2 point out that some logical classifiers like checking whether node is contained by a triangle can not be expressed by graded modal logic. It intrigues us to ask the following question:

1. Can we structure a new logic classifiers that is more expressive than graded modal logic?

2. What kind of GNNS is able to capture the new logic classifiers?

We will answers to the first question in the next subsections and the second question in the next two sections.

## 3.2 L-DIV GRADED LOGIC

Review the reason why GNNs fails in checking whether node is contained by a triangle, we found out the reason is that GNNs lack the ability to discriminate different neighbor. Hence we consider the formula with atomic formulas: $X = Y, E_{(0)}(x, y), E_{(1)}(x, y), label(x)$ as Boolean classifiers, and else same as logic $FO$, $E_{(0)}(x, y)$ is True iff $y \in N_{2_{(0)}}(x)$ and $E_{(1)}(x, y)$ is True iff $y \in N_{2_{(0)}}(x)$, we call the logic classifier as $2 - div\ FO$. Since $E(x, y) = E_{(0)}(x, y) \vee E_{(1)}(x, y)$, every formulas in $FO$ can be expressed by $2 - div\ FO$. For example,

$$\alpha(x) := \exists y[E(x, y) \wedge \exists z(E(z, x) \wedge E(y, z))] \tag{6}$$

Formula $\alpha(x)$ is evaluated to be true if node x is contained in a triangle in a undirected graph. We can express it by $2 - div\ FO$:

$$\alpha(x) := \exists y[(E_{(0)}(x, y) \vee E_{(1)}(x, y)) \wedge \exists (z(E_{(0)}(z, x) \vee E_{(1)}(z, x)) \wedge (E_{(0)}(y, z) \vee E_{(1)}(y, z)))] \tag{7}$$

There are three terms x,y,z in $\alpha(x)$, however it is possible to express an equivalent formula with fewer variables:

$$\alpha'(x) := \exists y(E_{(1)}(x, y)) \tag{8}$$

**Equation (8)** inspires us that some formulas that cannot be expressed by $FOC_2$ is able to be expressed by $2 - div\ FOC_2$. Hence it is possible to extend the logic of formula in $FOC_2$, that some formula can be expressed within less variables. Based on this idea, same as $2 - div\ FO$, replace atomic formulas with $X = Y, E_{l_{(i_2, i_3 \cdots i_l)}}(x, y), label(x)$, where $E_{(i_2, i_3 \cdots i_l)}(x, y)$ is $true$ iff $y \in N_{l_{(i_2 \cdots i_l)}}(x)$, and for the rest same as $FOC_2$, we denote the logic as $l - div\ FOC_2$.

We now can answwer the first question by introducing l-div graded modal logic, same as graded modal logic, the model for l-div graded modal logic is $M = (V, R_{l_{(i_2 \cdots i_l)}}, W)$, where V is a non-empty set of nodes, $R_{l_{(i_2 \cdots i_l)}}$ is a binary relation on V, in graph $xR_{l_{(i_2 \cdots i_l)}}y$ denotes $y \in N_{l_{(i_2 \cdots i_l)}}(X)$, and W is a valuation, which is a function mapping a subset of V to every letter. Graded modal formulas is formed with the atomic formula $W(v_1, v_2, \cdots v_n) = q$ and boolean connectives $\neg\ \wedge$, and the modal operators is defined as follow:

$$M, v \models \lozenge^n_{l_{(i_2 \cdots i_l)}} \varphi \iff \exists^{\neq} v_1 \ldots v_n \bigwedge_{1 \leq i \leq n} (vR_{l_{(i_2 \cdots i_l)}} v_i \wedge M, v_i \models \varphi) \tag{9}$$

**Theorem 3.1.** *If $l \leq 2$, every FO formula that can be expressed by (l+1)-div graded modal logic classifier can also be expressed by l-div graded modal logic classifier*

**Corollary 3.1.** *If $l \leq 2$, every formula FO that can be expressed by graded modal logic is able to be expressed by l-div graded modal logic.*

**Theorem 3.2.** *Each l-div graded modal logic classifier is captured by a l-div AC-GNN.*

---

**Algorithm 1** L-Division Color Refinement

---

**Input:** Directed graph $G = (V, E, X_V)$, deep of division: $l$, the number of iteration T
**Output:** Labels of nodes
**Initialize:** $c_v^{(0)} \leftarrow \text{hash}(X_v)(\forall v \in V), \quad t = 1$
**While** not converged or $t \leq T$
$c_v^t \leftarrow hash(c_v^{t-1}, \{\{c_w^{t-1} : w \in \mathcal{N}_{l_{(i_2, i_3 \cdots i_l)}}(v)\}\}), \forall v \in V, (i_2, i_3 \cdots i_l) \in (0,1)^{l-1}, t = t+1$

---

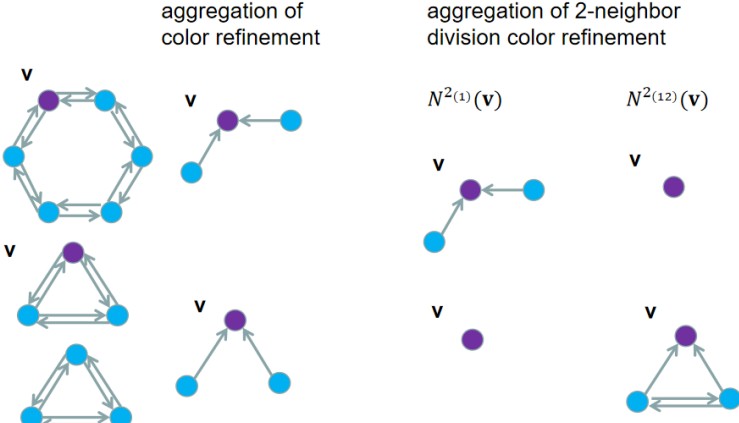

Figure 3: It shows the process for color refinement and 2-neighbor division color refinement. The color of color refinement, which is as powerful as 2-WL, operating on pair of two graphs for each node will always be the same, meaning that color refinement is not able to distinguish this pair of non-isomorphism graphs.While in 2-neighbor division color refinement, the neighbors of node v in two graphs are distributed into different multiset, the color for each graph will not be the same, meaning that 2-neighbor division color refinement is able to distinguish this pair of non-isomorphism graphs

## 4 NEIGHBOR L-DIVISION COLOR REFINEMENT ALGORITHM

Color refinement algorithm, also known as the 1-dimensional Weisfeiler-Leman algorithm, is widely used for detecting whether a pair of graphs are isomorphic. It starts with node's initial feature as node's color, then update the color by gathering information from node's neighbors, and combine them along with node's previous color, the algorithm stops when it convergences or reaches the largest iterations. $FOC_k$ is the fragment of FOC, which consists of all formulas that contain at most k distinct variables. Graph A,B is said as $C^k - equivalent$, denoted as $A \equiv_C^k B$, if any formula $\alpha \in FOC_K$, and any set of nodes $S_A$ in graph A that $|S_A| = k$, there exists set of nodes $S_B$ with $|S_B| = k$, that satisfies $\alpha(S_A) = \alpha(S_B)$. Color refinement algorithm is strongly connected with $FOC_2$,

**Proposition 3.** $(Immerman \ and \ Lander)$. For two graphs A and B, $A \equiv_C^2 B$ if, any only if, color refinement does not distinguish A and B.

**Proposition 4.** $(Cai, Furer, and Immerman)$. For two graphs A and B, $A \equiv_C^k B$ if, and only if, k-WL does not distinguish A and B.

We extend color refinement algorithm and propose l-div color refinement algorithm to enhance algorithm's expressive power in distinguishing non-isomorphism graphs. It discriminates node's neighbors by distributing different types of node's neighbors into different multisets, injective hash function makes it sure that node with different node's neighbors can be discriminated by the algorithm. Figure 3 shows that there's a pair of non-isomorphism graphs that 2-div color refinement algorithm is able to distinguish, while color refinement fails.

**Lemma 4.1.** For $l \geq 2$, there exists a pair of non-isomorphic graphs that l-division color refinement algorithm outputs "possibly isomorphic" while (l+1)-division color refinement algorithm outputs "non-isomorphic"

**Theorem 4.1.** *For $l \geq 2$, (l+1)-division color refinement algorithm is strictly more powerful than l-division color refinement algorithm.*

**Corollary 4.1.** *2-neighbor division color refinement algorithm is strictly more powerful than color refinement algorithm.*

## 5 NEIGHBOR DIVISION FRAMEWORK

In this section, we will answer the second question: What kind of GNNS is able to capture l-div graded logic classifiers. Denote $h_v^{(i)}$ as the representation of node v in layer i. An aggregate-combine GNN which follows neighborhood aggregation strategy, then update representation by combining neighborhood information and former representation as follow:.

$$m_v^{(i)} = AGGREGATE^{(i)} \left( \{\{h_u^{(i-1)} : u \in \mathcal{N}(v)\}\} \right) \tag{10}$$

$$h_v^{(i)} = \text{COMBINE}^{(i)} \left( h_v^{(i-1)}, m_v^{(i)} \right) \tag{11}$$

A common choice for aggregation function is to sum node's neighbors' representation, hence every iteration of AC-GNNs can be expressed as:

$$h_v^{(i)} = \sigma(h_v^{(i-1)} \cdot W_1^i + \sum_{u \in N(v)} h_u^{(i-1)} \cdot W_2^i + C^i) \tag{12}$$

Characteristically, we focus on how to enhance the expressive power more than 1-WL test. By reviewing the task of detecting triangle, the shortcoming of AC-GNNs is that them ignore node's neighbors' local structure information. A natural improvement for AC-GNNs is to classify node's neighbors into different categories. Formally, we propose L-division framework for AC-GNN, making the model competent in distinguishing node's neighbors with different local structure to enhance model's expressive power, the aggregation and combination function are as follow :

$$m_v^{(i,l_{(i_2\cdots i_l)})} = AGGREGATE^{(i)}(\{\{h_u^{(i-1)} : u \in N_{l_{(i_2\cdots i_l)}}(v)\}\}) \tag{13}$$

$$h_v^{(i)} = \text{COMBINE}^{(i)} \left( h_v^{(i-1)}, \{m_v^{(i,l_{(i_2\cdots i_l)})}\} \right) \quad (i_2 \cdots i_l) \in (0,1)^{l-1} \tag{14}$$

If the choice of aggregate and combination function is to sum up the representation in each category, the framework can also be expressed as:

$$h_v^{(i)} = \sigma(h_v^{(i-1)} \cdot W_1^i + \sum_{(i_2\cdots i_l)\in(0,1)^{l-1}} \sum_{u \in N_{l_{(i_2\cdots i_l)}}(v)} h_u^{(i-1)} \cdot W_{(i_2\cdots i_l)}^i + C^i) \tag{15}$$

To answer the second question, we still have to prove that l-div framework is able to capture l-div graded modal logic. We represent two defination to investigate their relationship.

**Definition 5.1.** *(Equivalent logic): We define the set of all logical classifiers that the GNN can captures as GNN's equivalent logic, denote as $tp(GNN)$.*

**Definition 5.2.** *For two $GNN_1$ and $GNN_2$, we say $GNN_2$ is logically more expressive than $GNN_1$, iff $tp(GNN_1) \subseteq tp(GNN_2)$.*

**Theorem 5.1.** *A logical classifier is captured by l-Div-AC-GNNs if and only if it can be expressed in l-div graded modal logic.*

Comparing with original AC-GNNs, it is appropriate for improved framework to be assumed that the ouput should not be *deviated* from the output by its original AC-GNNs. Therefore to describe this ability we define three properties that the framework should inherit

**Property 1. Invariance and Equivariance**: In graph-level task, permutation $\sigma$ for a AC-GNN is said to be invariant, if $\forall = (V, E, X)$ with $|V| = n$, denote A as indeuced adjacency matrix and $\sigma(G) = (\sigma(V), \sigma(E), \sigma(X))$ with indeuced adjacency matrix $\sigma(A)$, $AC - GNN(\sigma(G)) = AC -$

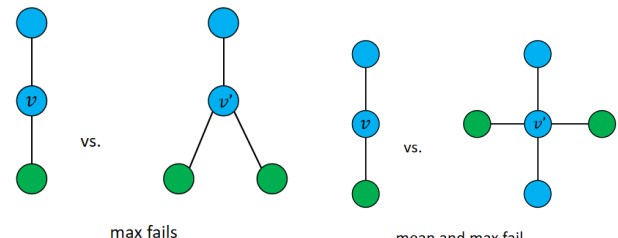

Figure 4: Given a formula as $\alpha(x) = \exists^{\geq 2} y(Green(y) \wedge E(x,y))$, max aggregator will fail in capturing $\alpha$ for the first pair of graphs. And mean and max will fail for the second pair of graphs

$GNN(G)$, the permutation set containing all such $\sigma$ is denoted as $S_I^{AC-GNN}(n)$. Same argument, In node-level task, permutation $\sigma$ for a AC-GNN is said to be equivariant, if $\forall = (V, E, X)$ with $|V| = n$, $AC - GNN(\sigma(G)) = \sigma(AC - GNN(G))$, the permutation set containing all such $\sigma$ is denoted as $S_E^{AC-GNN}(n)$. Denote the the permutation set of improved framewrok $AC - GNN_{imp}$ as $S_I^{AC-GNN_{imp}}(n)$ and $S_E^{AC-GNN_{imp}}(n)$. We say $AC - GNN_{imp}$ inherits invariance and equivariance, if $\forall n \in N, S_I^{AC-GNN}(n) \subseteq S_I^{AC-GNN_{imp}}(n), S_E^{AC-GNN}(n) \subseteq S_E^{AC-GNN_{imp}}(n)$

**Property 2. Approximate:** Given $\forall$ graph with feature $\vec{G} = (V, \vec{E}, X)$ and parameter setting $\Theta$ for acertain AC-GNN, denote the improved framework of AC-GNN as $AC - GNN_{imp}$, then there exists a parameter setting $\Theta_{imp}$ for $AC - GNN_{imp}$ that the output of AC-GNN is equal to $AC - GNN_{imp}$. Foemally, $\forall G = (V, E, X), \forall \Theta, \exists \Theta_{imp}, AC - GNN(G) = AC - GNN_{imp}(G)$.

**Property 3. Logic Expressive Power:** For $\forall$ AC-GNN, $tp(AC - GNN) \subseteq tp(AC - GNN_{imp})$

**Definition 5.3.** *For any set $X$ and $X' = X \bigcup x$ is a set and We say an AC-GNN is of countable additivity, if at every layer $i$ its aggregate function has the property as:*

$$AGGREGATE^{(i)}(X') = AGGREGATE^{(i)}(X) + AGGREGATE^{(i)}(\{x\}) \qquad (16)$$

**Theorem 5.2.** *Given AC-GNN of countable additivity, then its l-div framework inherits three Properties above.*

## 6 LOGIC EXPRESSIVE POWER OF AGGREGATORS

Figure 4 shows the logic power of aggregators as sum,mean and max, . Consider formula for node v $\varphi = \Diamond^{\geq 2}$, sum aggregator is able to capture the graded logic classifier. However, in figure 3 it shows that there are some graded logic classifier that mean and max aggregator. To explore the expressive power of GNNs which follows mean and max aggregator, we limit the modal operator in graded modal logic by replacing the counting quantifiers by quantifiers as:

$$M, w \models \Diamond' \varphi \iff \exists v(Rwv \wedge M, v \models \varphi) \qquad (17)$$

We denote the set of Logical classifier formula as $L'_{GML}$

**Theorem 6.1.** *Logical classifier $L'_{GML}$ can be captured by AC-GNNs which uses mean and max aggregator .*

## 7 EXPERIMENTAL RESULT

In this section, we utilize synthetic data to perform experiment to validates theorem that a for our theory. There are few questions that we wish to validate through experiments:

(1): Is l-div framework able to capture l-div graded modal logic?

(2): Is sum aggregator able to capture counting quantifier while sum $\exists^{\geq n}$ and does GNN with mean or max aggregator fail in capturing counting quantifier?

(3): Will the extra layers of GNN obstruct GNN's logic expressiveness.

**Logic Expressiveness Of L-Div GNN** The aim of this experiment is to show the expressiveness of 2-division GNN. We perform the experiment using common model GIN,GCN,GAT and their 2-div frameworks. We designed two task as:(1) detecting whether a node is contained by a triangle (2) counting the number of triangles that contains the node, to show that 2-div GNN is able to learn a very simple node classifier $\alpha$ which can be expressed by l-div graded modal logic while graded modal logic cannot. We generate two different type graph as erdos renyi graph and random regular graph with 4000,5000,6000 nodes. Table F shows the result of task 1: model GIN,GCN,GAT is not able to perfectly solve the task. However, their 2-div frameworks achieves $100\%$ accuracy in the task. It shows that 2-div GNN is able to capture the formula of task 1, which can be expressed by 2-div graded modal logic. Table F shows the result of task 2: even though task 2 cannot be expressed by 2-div graded modal logic the performance of GIN's,GCN's,GAT's 2-div frameworks are still better than themselves'. This experiment validates theorem 5.1.

**Logic Expressiveness Of aggregator** To answer the second question, we designed task 3 to detect whether node is contained by at least two triangles. The task is able to be expressed by 2-div graded modal logic but not by 2-div Logical classifier $2 - div\ L'_{GML}$. We used GIN with sum, mean and max aggregators to solve the task, Table F shows the result of task 3: aggregators mean and max is not able to perfectly solve the task and their : accuracy results are surprisingly concordant while sum aggregator achieves $100\%$ accuracy in the task 3. It validates that aggregators mean and max is not able to capture counting quantifier while sum aggregator is able to.

**Logic Expressiveness Of aggregator** To answer the third question, we use 2-div GIN with different layers to solve task 3. Table F shows the results: 2-div GIN achieves $100\%$ accuracy in the task 3 with number of layers 1,2,3. It shows the extra layers of GNN will not obstruct GNN's logic expressiveness. All the datas are provided in appendix.

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

# A INTRODUCTION OF MODAL THEORY

## A.1 MODAL LOGIC

We now use the following symbols to define terms and formulas:

(1) logical symbols: boolean connectives $\land, \lor, \neg, \to, \leftrightarrow$ as "and", 'or", "not", "implies" "iff" respectively. While every boolean connectives can be expressed by $\land, \lor, \neg, =$.

(2)the boolean quantifiers $\forall, \exists$ as "for all" and "there exists".

(3)constant symbols: usually denoted as c.

(4)function symbols: usually denoted as W with subscripts.

(5)relation symbols : usually denoted as R with subscripts.

(6)symbol = as and "equal"

**Definition A.1.** *A term is defined as follows:*

*(1)variable and constant are terms*

*(2)if W is an function symbol and $t_1 \cdots t_m$ are terms, then $W(t_1 \cdots t_m)$ is a term.*

**Definition A.2.** *A formula is defined as follows:*

*(1): if $t_1 \cdots t_m$ are terms and R is a relation symbols function, then $t_1 = t_2$ and $R(t_1 \cdots t_m)$ are formulas.*

*(2)If $\varphi$ and $\psi$ are formulas then $\varphi \land \psi, \varphi \lor \psi, \neg\psi, \varphi \to \psi, \varphi \leftrightarrow \psi$ are formulas.*

*(3)If x is a variable and $\psi$ is formulas, then $\forall x \psi(x)$ and $\exists x \psi(x)$ are formulas.*

For any terms $t_1 \cdots t_m$ and constant $C$, we regard $R(t_1 \cdots t_m)$,$t_1 = t_2$ and $W(t_1 \cdots t_m) = C$ as atomic formula.

**Proposition 5.** *Every formula in modal logic is formed by atomic formula and boolean connectives*

Model logic is formed with model and language which is a set of formulas. For graph, model is defined as follow:

**Definition A.3.** *In graph, A model (or structure) $\mathfrak{A}$ is usually formed as $< V, R, W >$, V is a nonempty set, usually denotes the set of graph nodes, R are relation symbol where for a graph R usually represents the information of graph edges set and W are function symbol which usually represents nodes' features.*

## A.2 GRADED MODAL THEORY

Graded modal formulas are usually formed with propositional variables $p, q, \ldots$, boolean connectives $\land, \neg$ and modal operators:$\Diamond^{\geq n}$ for $n \geq 1$. The language is usually denoted as $L_{GML}$.

# B METHOD TO MEASURE GNN'S EXPRESSIVENESS

Since XUer,(2019 a) propose that AC-GNN is *bound* by 1-WL, substantial frameworks have been proposed dedicating to enhance GNN's expressive power. However, there's still puzzlement that whether the framework is competent for a specific task. Sometime high expressive power framework has been choose which might cause high computational cost. To solve this problem, We now propose a framework to measure model's expressiveness: For given GNN, find its equivalent modal logic. For any given task, if the task is able to be transferred into GNN's equivalent modal logic then the GNN is competent for the task. Hence the key segment is to find model's equivalent modal logic. Here we roughly classify these improved framework into 3 categories and proposed a framework for their equivalent modal logic, please see appendix C for the details.

### B.1 EQUIVALENT MODAL LOGIC FOR HIGH-ORDER GNN

Based on high order k-WL or folklore k-WL tests, for $k \geq 2$, these GNNs regard k-tuple of nodes as a new unit node in the framework.(k-GNN's and local k-GNN's equivalent modal logics are provided in Appendix) The modals for these GNNs are usually as form:$M_{high-order} =< V^k, R_i, W >$, where $V^k$ usually is the k-tuple set, $R_i$ usually is a 2-placed relation on $V^k$. For example, in k-WL test, $R_i \boldsymbol{v}, \boldsymbol{\omega})(1 \geq i \geq k)$ is $true$ if $\boldsymbol{\omega} \in N_i(\boldsymbol{v})$ where $N_i(\boldsymbol{v}) = \{(v_1 \cdots, w, v_{i+1} \cdots v_k) | w \in V, w \neq v_i\}$ is $\boldsymbol{v's}$ i-th neighborhood. $W$ is an m-placed function that normally represents induced graph $G[\boldsymbol{v}]$ comparing with every contant graph set $H = \{h_t\}$ that $W(\boldsymbol{v}, H_t)$ is true iff $G[\boldsymbol{v}] = H_t$. Modal logic formula on $M_{high-order}$ are normally form with atomic formulas $W(\boldsymbol{v}, H_t)$, boolean connectives $\wedge, \neg$, and graded modal operator

$$M, \boldsymbol{v} \models \Diamond_i^n \varphi \iff \exists^{\neq} \boldsymbol{v_1} \ldots \boldsymbol{v_n} \bigwedge_{1 \leq l \leq n} (R(\boldsymbol{v}, \boldsymbol{v_l})) \wedge M, \boldsymbol{v_l} \models \varphi) \tag{18}$$

### B.2 EQUIVALENT MODAL LOGIC FOR SUBSTRUCTURE-BASED

Substructure-based GNNs usually enhance expressive power by appending node's local substructure information (Graph Substructure Networks equivalent modal logics are provided in Appendix) Given a graph G, the modals for these GNNs are usually as form:$M_{sub} =< V, R, W >$, where $V$ usually is the node set, $R$ usually is a 2-placed relation on $V$, $R(V, \Omega)$ is $true$ iff $(v, \omega) \in E$. $W$ is an m-placed function that normally represents node's local substructure information comparing with selected constant graph set $H = \{h_t\}$ that $W(v, H_t)$ is $true$ iff node v is contained in $H_t$ or $W$ has form as $W(v, H_t, p)$ where p is a constant, $W(v, H_t, p)$ is $true$ iff the number of different nodes set $S_i$ while $v \in S_i$ and induced graph $G[S_i]$ is isomorphic to $H_t$ is p. Modal logic formula on $M_{substructure-based}$ are normally formed with atomic formulas $W(\boldsymbol{v}, H_t)$ or $W(v, H_t, p)$, boolean connectives $\wedge, \neg$, and graded modal operator

$$M, v \models \Diamond^n \varphi \iff \exists^{\neq} v_1 \ldots v_n \bigwedge_{1 \leq l \leq n} (R_i(v, v_l)) \wedge M, v_l \models \varphi) \tag{19}$$

### B.3 EQUIVALENT MODAL LOGIC FOR GRAPH TRANSFORMATION

Graph transformation GNNs usually enhance expressive power by transforming graph into a new graph by predefination and caculate the output by aggregating across all the transformed graphs. ( Equivariant Subgraph Aggregation Networks modal logics are provided in Appendix) Given a graph G, the modals for these GNNs are usually as form:$M_{trans} =< V, R_i >$ and transforming policy $\pi_i(1 \geq i \geq k)$, where $V$ usually is the node set. Denote the identical transformation policy as $\pi_0, R_i$ usually is a 2-placed relation on $V$, $R_i(V, \Omega)(0 \geq i \geq k)$ is $true$ iff node $v$ and $\omega$ is connected in graph $\pi_i(G)$. Modal logic formula on $M_{trans}$ are normally formed with boolean connectives $\wedge, \neg$, and graded modal operator

$$M, v \models \Diamond_i^n \varphi \iff \exists^{\neq} v_1 \ldots v_n \bigwedge_{1 \leq l \leq n} (R_i(v, v_l)) \wedge M, v_l \models \varphi) \tag{20}$$

## C WEISFEILER-LEHMAN TEST AND MODAL LOGIC FOR RECENTLY PROPOSED VARIANTS

### C.1 1-DIMENSIONAL WEISFEILER-LEHMAN TEST(1-WL)

Given two graph $G = (V_1, E_1)$ and $H = (V_2, E_2)$, $G$ and $H$ are considered to be isomorphic if there is a bijection $\varphi$ between $V_1$ and $V_2$ that preserves nodes' adjacencies: $(E_1(v_1, \omega_1)$ if and only if $E_2(\varphi(v_1), \varphi(\omega_1)))$. 1-dimensional Weisfeiler-Lehman test is a algorithm calculating color for each node in $G$ and $H$. If there is a bijection between the color of two graphs. The algorithm will consider two graphs to be isomorphic. First, the color of each node is initialized to be the degree of node. Then, for each iteration of the algorithm, a tuple of color that containing node's predecessor color and the multiset of the node's neighbors' predecessor color will be compressed by a hash function.

Let $S_v^T$ be the node set of graph $G$ that the distance from node v is at most t and $G(S_v^T)$ be a induced graph by node set $S_v^T$. Notice if the largest number of iteration is T, then the color of

---

**Algorithm 2** 1-dimensional Weisfeiler-Lehman test(1-WL)

---

**Input:** graph $G = (V, E)$, the largest number of iteration: $T$
**Output:** color of nodes
**Initialize:**
$c_v^{(0)} \leftarrow \text{hash}(degree(v))(\forall v \in V)$
**While:** not converged or number of iteration t ≤ T
$c_v^t \leftarrow \text{hash}(c_v^{t-1}, \{\{c_w^{t-1} : w \in \mathcal{N}(v)\}\}) \ \forall v \in V$

---

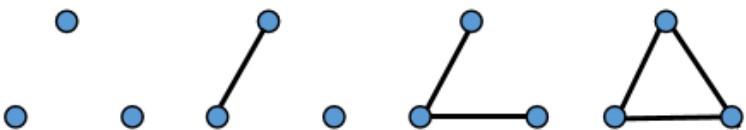

Figure 5: 4 different initialized color in 3-dimensional Weisfeiler-Lehman test

node v is determined by induced graph $G(S_v^T)$. Hence for every connected graph, there exists T that $G[S_v^T] = G$ for any node V. This implies 1-dimensional Weisfeiler-Lehman test will finally converge for every connected graph. If there's not a bijection between color of two graphs, then two graphs are definitively not isomorphic. However, there exists a pair of non-isomorphic graphs that share the same canonical form, however, the algorithm will determine two graphs are isomorphic.

### C.2 K-DIMENSIONAL FRAMEWWORK

#### C.2.1 K-DIMENSIONAL WEISFEILER-LEHMAN TEST

Given a graph $G = (V, E)$, k-dimensional Weisfeiler-Lehman test regrads $\boldsymbol{v} = (v_1, \cdots v_k) \in V^k$ as a k-tuple. Define $\vec{v}(i, \omega) \in V^k$ as a k-tuple that replaces $v_i$ by $\omega$ and $v[\vec{i, \omega}] \in V^k$ is said to be $\vec{v}$'s i-neighbor denoted as $\vec{v}[i, \omega] \in N_i(\vec{v})$. In k-dimensional Weisfeiler-Lehman test, every k-tuple's initialized color is based on the induced graph $G[v_1, \cdots v_k]$. Specifically two k-tuples is initialized with the same color if and only if they have same induced graph $G[v_1, \cdots v_k]$. For example, as the figure 5 shows, there are 4 different initialized color in 3-dimensional Weisfeiler-Lehman test.

#### C.2.2 K-DIMENSIONAL GRAPH NEURAL NETWORKS

Morris et al.(2019) proposed the framework of k-dimensional Graph Neural Networks based on the k-dimensional Weisfeiler-Lehman test(k-WL). The model considers every k-element subset $\vec{v} \in V(G)^k$ over node set $V(G)$. Due to restriction of computation complexity and GPU memory, $\vec{v}$'s i-neighbors $N_i(\vec{v})$ are combined into one neighbor set as:

$$N(\vec{v}) = \{\vec{\omega} \in [V(G)]^k \mid |\vec{v} \cap \vec{\omega}| = k - 1\} \tag{21}$$

If $\vec{\omega} \in N(\vec{v})$, let $\vec{\omega}/\vec{v}$ denotes the unique node $t$ that $t \in \vec{\omega}, t \notin \vec{v}$. The local neighborhood is defined as $N_L(\vec{v}) = \{\vec{\omega}|\vec{\omega} \in N(\vec{v}), (\vec{\omega}/\vec{v}, \vec{v}/\vec{\omega}) \in E(G)\}$ and global neighborhood $N_G(\vec{v}) = N(\vec{v})/N_L(\vec{v})$.

Denote $\chi_k^t(\vec{v})$ as the feature for tuple $\vec{v}$ at layer t. The feature $\chi_k^0(\vec{v})$ is initialized by a one-hot encoding feature function of the induced graph $G[\vec{v}]$ with input features. In k-GNN $\chi^t(\vec{v})$ is computed by:

$$\chi_k^t(\vec{v}) = \sigma(\chi_k^{t-1}(\vec{v}) \cdot W_1^{(t)} + \sum_{\omega \in N_L(\vec{v}) \cup N_G(\vec{v})} \chi_k^{t-1}(\vec{\omega}) \cdot W_2^{(t)} + C^t) \tag{22}$$

To prevent overfitting, Morris et al.(2019) proposed the local k-GNN as:

$$\chi_k^t(\vec{v}) = \sigma(\chi_k^{t-1}(\vec{v}) \cdot W_1^{(t)} + \sum_{\vec{\omega} \in N_L(\vec{v})} \chi_k^{t-1}(\vec{\omega}) \cdot W_2^{(t)} + C^t) \tag{23}$$

---

**Algorithm 3** k-dimensional Weisfeiler-Lehman test(k-WL)

---

**Input:** graph $G = (V, E)$, the largest number of iteration: $T$
**Output:** color of k-tuple $V^k$
**Initialize:**
$c_{\vec{v}}^{(0)} \leftarrow hash(G[v_1, \cdots v_k])(\forall \vec{v} \in V^k)$
**While:** not converged or the number of iteration t $\leq$ T
$c_{\vec{v}}^t \leftarrow hash(c_{\vec{v}}^{t-1}, \{\{c_{\vec{w}}^{t-1} : \vec{w} \in N_i(\vec{v})\}\}) \ \forall \vec{v} \in N^k$

---

### C.2.3 EQUIVALENT MODAL LOGIC

Given a graph G, we define the modal induced by k-GNN as $M_{k-GNN} = <$ $V^k, E_k, F_{k-GNN}, \{H^k\} >$, $V^k$ is the k-tuple set, $E_k$ is a 2-placed relation on $V^k$ and $E_k(\vec{v}, \vec{\omega})$ is $true$ iff $|\vec{v} \cap \vec{\omega}| = k-1$, $F_{k-GNN}$ is an 2-placed function that $F_{k-GNN}(\vec{v}, H^k)$ is true iff $G[\vec{v}] = H^k$, $\{H_k\}$ is a set of constant k nodes graph corresponding to $\{G[\vec{v}]|\vec{v} \in V^k\}$. The language is Graded modal formulas $L_{GML}$, with atomic formulas: $F_{k-GNN}(\vec{v}, H^k)$. And modal operatior $\Diamond$ which is defined as :

$$M, \vec{v} \models \Diamond^n \varphi \iff \exists^{\neq} \vec{v_1} \cdots \vec{v_n} \bigwedge_{1 \leqslant i \leqslant n} (E(\vec{v}, \vec{v_i})) \wedge M, \vec{v_i} \models \varphi) \tag{24}$$

Same argument, we define the modal induced by local k-GNN as $M_{k-GNN} = <$ $V^k, E_{k-local}, F_{k-GNN}, \{H_k\} >$, except for $E_{k-local}(\vec{v}, \vec{\omega})$ is $true$ iff $\vec{v} \cap \vec{\omega}| = k-1$ and $(\vec{\omega}/\vec{v}, \vec{v}/\vec{\omega}) \in E(G)$, the rest is same as k-GNN logic.

**Theorem C.1.** *k-GNN is able to capture k-GNN logic, local k-GNN is able to capture local k-GNN logic.*

*Proof.* For every k-GNN formula $\varphi(\vec{v})$, let $sub(\varphi) = (\varphi_1, \cdots \varphi_n)$ be a sequences of subformula of $\varphi(\boldsymbol{v})$ that every subformula $\vec{v})_i$ is formed as atomic formula: $F(\vec{V}) = H_i^k$ or like $\varphi_i = \varphi_j \wedge \varphi_l$, $\varphi_i = \neg \varphi_j$ or $\varphi_i = \Diamond^{\geq N} \varphi_j$ for $j, l < i$ and $\varphi_n = \varphi$. Formally construct a k-GNN or local k-GNN with n layers and the aggregation and combine function is as equation **??** or **??**, $W_1^t$ and $W_2^t$ are two $n \times n$ matrixs and $C^t$ is a $n - length$ vector. For $1 \leq t \leq n$, every layer are defined as follow:

if $\varphi_t = F_{k-GNN}(\vec{v}, H^k)$ , then let $W_1^t(t, t) = 1$

if $\varphi_t = \varphi_j \wedge \varphi_k$, then let $W_1^t(j, t) = 1, W_1^t(k, t) = 1$ and $C^t(i) = -1$

if $\gamma_t(x) = \neg \gamma_k$, then let $W_1^t(j, t) = -1$ and $C^t(i) = -1$

if $\gamma_t(x) = \Diamond^{\geq N} \gamma_j$, then let $W_2^t(j, t) = 1$ and $C^t(i) = N - 1$

and all the rest of values in the 't-th iteration of $W_1^t, W_2^t, C^t$ are 0. Activation function $\sigma(x) = min(max(0, x), 1)$. Then it is easy to prove such k-GNN and local k-GNN is able to capture formula $\varphi(\vec{v})$.

$\square$

**Theorem C.2.** *3-GNN and local 3-GNN is competent for detecting triangle task.*

*Proof.* Let $H^0$ be a complete graph with 3 nodes and $\varphi(\vec{v}) = (F(\vec{v}) = H^0)$. Notice $\varphi(\vec{v})$ can be expressed by 3-GNN and local 3-GNN logic and let $\alpha(v_0) = \exists_{\vec{v} \in V^k, v_0 \in \vec{v}}(\varphi(\vec{v}))$. Hence $\alpha(v_0)$ is $true$ iff node $v_0$ is contained by a triangle. $\square$

### C.3 GRAPH SUBSTRUCTURE NETWORKS(GSN)

Bouritsas et al. (2022) proposed a variant framework based on WL as Graph Substructure Networks(GSN) that is able to capture the structure of the underlying graph. To take the framework into application, first we need to specify a set of (small) selected connected graphs $H = \{H_1, \cdots H_k\}$. Given a graph $G = (\mathcal{V}, E)$, the extra node feature are defined as follow:

$$x_{H_i}^V(v) := |\{G[\mathcal{S}] : \mathcal{S} \subset \mathcal{V}, G[\mathcal{S}] \simeq H_i, v \in \mathcal{S}\}|, \quad i \in [d_H]. \tag{25}$$

$x_{H_i}^V(v)$ represents the number of induced graphs that contain node v while are isomorphism to $H_i$. $x_{H,i}^V(v)$ are combined into a vector $\boldsymbol{x^V(v)} = [x_{H_1}^V(v), \cdots x_{H_k}^V(v)]$ as node's extra Substructure feature. Denote $\chi_G^{t-1}(v)$ as node $v's$ feature at layer t, the aggregation and combine function can be denoted as follow:

$$\chi_G^t(v) := \sigma\left((\chi_G^{t-1}(v), \boldsymbol{x}^V(v)) \cdot W_1^t, \sum_{u \in \mathcal{N}_G(v)} (\chi_G^{t-1}(u), \boldsymbol{x}^V(u)) \cdot W_2^t + C^t\right) \quad (26)$$

### C.3.1 EQUIVALENT MODAL LOGIC

Given a graph G, we define the modal induced by Graph Substructure Networks as $M_{GSN} =< V, E, F_{GSN}, \{H_i\}, N >$, $V$ is the node set, $E$ is a 2-placed relation on $V$ and for $E(v, \omega)$ is $true$ iff $(v, \omega) \in E$, $F_{GSN}$ is an 2-placed function that $F_{GSN}(v, H_i) = x_{H_i}^{(}v)$, $\{H_k\}$ is a set of constant graphs corresponding to the selected connected graphs in Graph Substructure Networks framework. The language is Graded modal formulas $L_{GML}$, with atomic formulas: $F_{GSN}(v, H_i) \geq q(q \in N)$. The modal operatior is defined as :

$$M, v \models \Diamond^n \varphi \iff \exists^{\neq} v_1 \ldots v_n \bigwedge_{1 \leqslant i \leqslant n} (E(v, v_i)) \land M, v_i \models \varphi) \quad (27)$$

**Theorem C.3.** *Graph Substructure Networks is able to capture Graph Substructure Networks logic.*

*Proof.* For every Graph Substructure Networks formula $\varphi(v)$, let $sub(\varphi) = (\varphi_1, \cdots \varphi_n)$ be a sequences of subformula of $\varphi(v)$ that every subformula $_i$ is formed as atomic formula: $v = \omega$, $F_{GSN}(v, H_i) = q$ or like $\varphi_i = \varphi_j \land \varphi_l$, $\varphi_i = \neg \varphi_j$ or $\varphi_i = \Diamond^{\geq N} \varphi_j$ for $j, l < i$ and $\varphi_n = \varphi$. Formally construct a Graph Substructure Network with n layers and the aggregation and combine function is as equation ,$W_1^t$ and $W_2^t$ are two $(n + k) \times n$ matrices and $C^t$ is a $n - length$ vector. For $1 \leq t \leq n$, every layer are defined as follow:

if $\varphi_t = (F_{GSN}(v, H_i) \geq q)$ then let $W_1^t(i + n, t) = 1$ and $C^t(i) = -q + 1$

if $\varphi_t = \varphi_j \land \varphi_k$, then let $W_1^t(j, t) = 1$, $W_1^t(k, t) = 1$ and $C^t(i) = -1$

if $\varphi_t = \neg \varphi_k$, then let $W_1^t(k, t) = -1$ and $C^t(t) = -1$

if $\varphi_t = \Diamond^{\geq N} \varphi_j$, then let $W_2^t(j, t) = 1$ and $C^t(i) = -N + 1$

and all the rest of values in the 't-th iteration of $W_1^t, W_2^t, C^t$ are 0. Activation function $\sigma(x) = min(max(0, x), 1)$. Then it is easy to prove such Graph Substructure Networks is able to capture formula $\varphi(\boldsymbol{v})$.

$\square$

**Theorem C.4.** *Graph Substructure Networks is competent for detecting triangle task.*

*Proof.* Let $H_0$ be a complete graph with 3 nodes and $\varphi(\boldsymbol{v}) = (F_{GSN}(v, H_0) \geq 1)$. Notice $\varphi(v)$ can be expressed by Graph Substructure Networks logic. Hence $\varphi(v)$ is $true$ iff node $v$ is contained by a triangle $\square$

### C.4 EQUIVARIANT SUBGRAPH AGGREGATION NETWORKS (ESAN)

Bevilacqua et al. (2022) proposed a new framework of graph neural networks, called as Equivariant Subgraph Aggregation Networks. Based on the predefined policy $\pi$, the network first generates a set of graphs $\mathcal{B}_G^\pi = \{\{G_1, \cdots, G_m\}\}$ given a graph $G = (V, E)$. Every $G_i = (V, E_i)$ in $\mathcal{B}_G$ shares the same node set $V$, but different in edges set $E_i$. The initial color $\chi_{G_i}^0(v)$ for every node is based on policy $\pi$ and input features. The aggregation and combine function can be expressed as follow:

$$\chi_{G_i}^t(v) = \sigma_1\left(\chi_{G_i}^{t-1}(v) \cdot W_{(1,i)}^t + \sum_{u \in \mathcal{N}_{G_i}(v)} \chi_{G_i}^{t-1}(u) \cdot W_{(2,i)}^t + \chi_G^{t-1}(v) \cdot W_1^t + \sum_{u \in \mathcal{N}_G(v)} \chi_G^{t-1}(u) \cdot W_2^t + C_i^t\right)$$
$$(28)$$
$$\chi_G^t(v) = \sigma_2(\chi_{G_i}^t(v)) \quad (29)$$

### C.4.1 EQUIVALENT MODAL LOGIC

Given a graph G, we define the modal induced by Equivariant Subgraph Aggregation Networks as $M_{ESAN} = <V, E, \pi_i>$, $V$ is the node set, $\pi_i$ is an 1-placed function that map the node $v \in G$ to the corresponding node in graph $G_i$. $\pi_0$ is define as identical function as map node $v \in G$ to itself, let $G_0 = \pi_0(G) = G$. $E$ is a 2-placed relation on $V$ and for $0 \leq i \leq m$, $E(\pi_i(v), \pi_i(\omega))$ is $true$ iff $(\pi_i(v), \pi_i(\omega)) \in E_i$.

The equivalent language formulas $L_{ESAN}$ are built up using boolean connectives $\neg, \wedge$, and the modal operators $\Diamond^{n,i}$ is defined as:

$$\text{M,v} \models \Diamond^{n,i}\{\varphi\} \iff \exists^{\neq} v_1 \ldots v_n \bigwedge_{1 \leqslant j \leqslant n} (E(\pi_i(v), \pi_i(v_j)) \wedge M, \pi_i(v_j) \models \varphi) \quad (30)$$

**Theorem C.5.** *Equivariant Subgraph Aggregation Networks is able to capture Equivariant Subgraph Aggregation logic.*

*Proof.* For every Equivariant Subgraph Aggregation formula $\varphi(v)$, let $sub(\varphi) = (\varphi_1, \cdots \varphi_n)$ be a sequences of subformula of $\varphi(v)$ that every subformula $_i$ is formed as formula: $\varphi_t = \varphi_j \wedge \varphi_l$, $\varphi_t = \neg\varphi_j$ or $\varphi_t = \Diamond^{n,i}\varphi_j$ for $j, l < i$ and $\varphi_n = \varphi$. Formally construct a Graph Substructure Network with n layers and the aggregation and combine function is as equation **??**, $W_{(1,i)}^t$, $W_{(2,i)}^t$, $W_1^t$ and $W_2^t$ are $n \times n$ matrixs and $C_i^t$ are $n - length$ vectors. For $1 \leq t \leq n$, every layer are defined as follow:

if $\varphi_t = \varphi_l \wedge \varphi_k$, then let $W_1^t(k,t) = 1$, $W_1^t(l,t) = 1$ and $C_i^t(t) = -1$

if $\varphi_t = \neg\varphi_k$, then let $W_1^t(k,t) = -1$ and $C^t(t) = -1$

if $\varphi_t = \Diamond^{n,i}\{\varphi_l\}$ and $i = 0$, then let $W_2^t(l,t) = 1$ and $C_j^t(t) = -n+1$ for $1 \geq j \geq m$

if $\varphi_t = \Diamond^{n,i}\{\varphi_l\}$ and $i \neq 0$, then let $W_{(2,i)}^t(l,t) = 1$ and $C_i^t(t) = -n+1$

and all the rest of values in the 't-th iteration of $W_{(1,i)}^t$, $W_{(2,i)}^t$, $W_1^t$ and $W_2^t$ are 0. Activation function $\sigma_1(x) = min(max(0,x),1)$, $\sigma_2(x) = max(x)$. Then it is easy to prove such Graph Substructure Networks is able to capture formula $\varphi(\boldsymbol{v})$. $\square$

**Theorem C.6.** *Equivariant Subgraph Aggregation Networks is competent for detecting triangle task.*

*Proof.* Notice if node $\omega$ is node v's $2_{(1)}$ neighbor, node $v$ is node $\omega$'s $2_{(1)}$ Neighbor. Define the policy $\pi_1$ as follow: if node $\omega$ is node v's $2_{(1)}$ neighbor, $(\pi_1(v), \pi_1(\omega)) \in E_1$. By theorem for 2-div network, Equivariant Subgraph Aggregation Networks is competent for detecting triangle task. $\square$

### C.5 SHORTEST PATH DISTANCE WL(SPD-WL)

SPD-WL is variant version of DSS-WL designed to solve biconnectivity problems. Given a graph $G = (V, E)$, SPD-WL chooses the shortest path distance as the policy. The aggregation function of SPD-WL algorithm can be expressed as:

$$\chi_G^t(v) := \text{hash}(\chi_G^{t-1}(v), \{\{\chi_G^{t-1}(u) : u \in \mathcal{N}_G(v)\}, \{\{\chi_G^{t-1}(u) : \text{dis}_G(v,u) = 2\}\}, \cdots, \{\{\chi_G^{t-1}(u) : \text{dis}_G(v,u) = n-1\}\}, \{\{\chi_G^{t-1}(u) : \text{dis}_G(v,u) = \infty\}\}). \quad (31)$$

### C.5.1 EQUIVALENT MODAL LOGIC

Given a graph G, we define the modal induced by Shortest Path Distance WL as $M_{ESAN} = <V, \{DIS_i\}>$. $V$ is the node set, $DIS_i$ is a 2-placed relation on $V$ and $DIS_i(v, \omega)$ is $true$ iff the distance of node v and $\omega$ equals to $i$.

The equivalent language formulas $L_{ESAN}$ are built up using boolean connectives $\neg, \wedge$, and the modal operators $\Diamond^{n,i}$ is defined as:

$$\text{M,v} \models \Diamond^{n,i}\{\varphi\} \iff \exists^{\neq} v_1 \ldots v_n \bigwedge_{1 \leqslant j \leqslant n}(DIS_i(v, v_j) \wedge M, v_j \models \varphi) \tag{32}$$

**Theorem C.7.** *Equivariant Subgraph Aggregation Networks is able to capture Equivariant Subgraph Aggregation logic.*

*Proof.* The Shortest Path Distance network can be expressed as:

$$\chi_G^t(v) := \sigma(\chi_G^{t-1}(v) \cdot W_1^t + \sum_i \sum_{dis_G(v,u)=i} \chi_G^{t-1}(u) \cdot W_{2,i}^t + C^t). \tag{33}$$

For every Shortest Path Distance network formula $\varphi(v)$, let $sub(\varphi) = (\varphi_1, \cdots \varphi_n)$ be a sequences of subformula of $\varphi(v)$ that every subformula $_i$ is formed as formula: $\varphi_t = \varphi_j \wedge \varphi_l$, $\varphi_t = \neg\varphi_j$ or $\varphi_t = \Diamond^{n,i}\varphi_j$ for $j, l < i$ and $\varphi_n = \varphi$. Formally construct a Shortest Path Distance Network with n layers and the aggregation and combine function is as equation , $W_1^t$, $W_{(2,i)}^t$ are $n \times n$ matrixs and $C^t$ are $n - length$ vectors. For $1 \leq t \leq n$, every layer are defined as follow:

if $\varphi_t = \varphi_l \wedge \varphi_k$, then let $W_1^t(k,t) = 1$, $W_1^t(l,t) = 1$ and $C_i^t(t) = -1$

if $\varphi_t = \neg\varphi_k$, then let $W_1^t(k,t) = -1$ and $C^t(t) = -1$

if $\varphi_t = \Diamond^{n,i}\{\varphi_l\}$, then let $W_{(2,i)}^t(l,t) = 1$ and $C^t(t) = -n + 1$ for $1 \geq j \geq m$

and all the rest of values in the 't-th iteration of $W_1^t$, $W_{(2,i)}^t$ and $C^t$ are 0. Activation function $\sigma_1(x) = min(max(0, x), 1)$. Then it is easy to prove such Graph Substructure Networks is able to capture formula $\varphi(\boldsymbol{v})$. $\qquad\square$

# D   PROOFS

In this section we will provide all the proof of lemmas, theorems and propositions. We will restate them for the convenience.

## D.1   PROOFS OF HIERARCHY OF NODE'S NEIGHBOR CLASSIFICATION

**Lemma D.1.** *Let $\vec{G}$ be a graph with adjacency matrix A. The number of walks from u to v in $\vec{G}$ with length k is $(A^k)_{u,v}$*

*Proof.* When K=1, $(A^k)$ is adjacency matrix A, hence $(A)_{u,v}$ represents the number of edges from node u to node v. lemma 1 holds when k=1.

Assume lemma 1 holds when K=k, $(A)_{u,v}^k$ represents the number of walks from u to v in $\vec{G}$ with length k, since the number of walks from u to v in $\vec{G}$ with length k+1 equals to the sum of the number of edges $(\vec{u,i})$, for all node i, multiplies the number of walks from i to v which is $\sum_{i=1}^{|V|} a_{u,i} \cdot (A)_{i,v}^k$. Note that $\sum_{i=1}^{|V|} a_{u,i} \cdot (A)_{i,v}^k = (A)^{(}k+1)_{u,v}$, hence lemma 1 holds when K=k+1. $\qquad\square$

**Theorem D.1.** *Given sequence $(i_{(k+1)}, i_{(k+2)} \cdots, i_l) \in (0,1)^{(l-k)}$, a graph $G = (V, E)$ with induced adjacency matrix A and $t(k + 1 \leq t \leq l)$, define $(A^t)^{(i_{(k+1)} \cdots, i_l)} = A^t$ if $i_t = 1$ , else $(A^t)^{(i_{(k+1)} \cdots, i_l)} = \sim A^t$ if $i_t = 0$, then $A_{L_{(i_{(k+1)} \cdots, i_l)}}^K = Mask_{\prod_{t=K+1}^L \odot (A^t)^{(i_{(k+1)} \cdots, i_l)}}(A^K)$*

*Proof.* When L=K, $Mask_{\prod_{t=k+1}^L \odot (A^t)^{i_{(k+1)}, i_{(k+2)} \cdots i_L}}(\cdot)$ is identical operator, while $A_{L_{i_1, i_2 \cdots i_t}}^k = A^K$ is K-hop adjacency matrix, so theorem holds when L=K.

Assume theorem 1 holds when L=l(l¿k), that is $A^k_{l_{i_{(k+1)},i_{(k+2)}\cdots i_L}} = Mask_{\prod^l_{t=k+1} \odot (A^t)^{i_1,i_2\cdots,i_l}}(A_k)(K = i_1 < i_2 < \cdots < i_t \leq l)$. Hence if $(Mask_{\prod^l_{t=k+1} \odot (A^t)^{i_1,i_2\cdots,i_l}}(A_k)(K = i_1 < i_2 < \cdots < i_t \leq l))_{(u,v)} = 1$, then $u \in N^{L_{(i_{(K+1)},i_{(K+2)}\cdots i_l)}}_{k-hop}(v)$, else $u \notin N^{L_{(i_{(K+1)},i_{(K+2)}\cdots i_l)}}_{k-hop}(v)$

Consider when L=l+1(l¿k), if $i_{(l+1)} = 0$, $(A^k_{l_{i_{(k+1)},i_{(k+2)}\cdots i_{(l+1)}}})_{(u,v)} = 1$ implies $u \in N^{L_{(i_{(K+1)},i_{(K+2)}\cdots i_{(l+1)})}}_{k-hop}(v)$ , while as lemma 1 indicates $(A^{(l+1)})_{u,v}$ the number of walks from node u to node v with length l+1, then there does not exist walk from node u to node v with length l+1, hence if $(A^{(l+1)})_{u,v} = 0$ then there does not exist any directed walk from node u to node v with length l+1, notice that operator

$$Mask_{\prod^{l+1}_{t=k+1} \odot (A^t)^{i_1,i_2\cdots,i_{(l+1)}}}(\cdot) = Mask_{(A^{l+1})^{i_1,i_2\cdots,i_{(l+1)}}}(Mask_{\prod^l_{t=k+1} \odot (A^t)^{i_1,i_2\cdots,i_l}}(\cdot)) \tag{34}$$

and

$$(Mask_{(A^{l+1})^{i_1,i_2\cdots,i_{(l+1)}}}(A^k_{l_{i_{(k+1)},i_{(k+2)}\cdots i_{(l)}}}))_{(u,v)} = (A^k_{l_{i_{(k+1)},i_{(k+2)}\cdots i_{(l)}}})_{(u,v)} \tag{35}$$

if $(A^{(l+1)})_{(u,v)} = 0$, which implies that $u \in N^{L_{(i_{(K+1)},i_{(K+2)}\cdots i_{(l+1)})}}_{k-hop}(v)$. Else $(Mask_{(A^{l+1})^{i_1,i_2\cdots,i_{(l+1)}}}(A^k_{l_{i_{(k+1)},i_{(k+2)}\cdots i_{(l)}}}))_{(u,v)} = 0$, if $(A^{(l+1)})_{(u,v)} > 0$, which also means that $u \in N^{L_{(i_{(K+1)},i_{(K+2)}\cdots i_{(l+1)})}}_{k-hop}(v)$. Hence theorem 1 holds when $i_{(l+1)} = 0$.

Same argument, if $i_{(l+1)} = 1$, then $(A^k_{l_{i_{(k+1)},i_{(k+2)}\cdots i_{(l+1)}}})_{(u,v)} > 0$ means there does exists walk from node u to node v with length l+1, hence if $(A^{(l+1)})_{u,v} = 0$ then there does not exist any directed walk from node u to node v with length l+1, and $(Mask_{(A^{l+1})^{i_1,i_2\cdots,i_{(l+1)}}}(A^k_{l_{i_{(k+1)},i_{(k+2)}\cdots i_{(l)}}}))_{(u,v)} = (A^k_{l_{i_{(k+1)},i_{(k+2)}\cdots i_{(l)}}})_{(u,v)}$,if $(A^{(l+1)})_{(u,v)} = 1$, implies that $u \in N^{L_{(i_{(K+1)},i_{(K+2)}\cdots i_{(l+1)})}}_{k-hop}(v)$. Else $(Mask_{(A^{l+1})^{i_1,i_2\cdots,i_{(l+1)}}}(A^k_{l_{i_{(k+1)},i_{(k+2)}\cdots i_{(l)}}}))_{(u,v)} = 0$, if $(A^{(l+1)})_{(u,v)} = 0$, which also means that $u \in N^{L_{(i_{(K+1)},i_{(K+2)}\cdots i_{(l+1)})}}_{k-hop}(v)$.Hence theorem 1 holds when $L = l(l > K)$. $\square$

## D.2 PROOFS OF L-DIV GRADED MODEL LOGIC

**Theorem D.2.** *If $l \geq 2$, every FO formula that can be expressed by (l+1)-div graded modal logic classifier can also be expressed by l-div graded modal logic classifier*

*Proof.* Given l-div graded modal logic formula $\varphi$, let $sub(\varphi) = (\varphi_1, \cdots, \varphi_N)$ be enumeration of sub-formulas where $\varphi_i$ is formed by $\varphi_j$ and connectives for $j < i$ and $\varphi_N = \varphi$. Assume theorem holds for every formula when $N \leq n$, for every formula that its enumeration of sub-formulas is $(\varphi_1, \cdots, \varphi_n, \varphi_{n+1})$:

If $\varphi_{N+1} = \neg\varphi_i$ or $\varphi_{n+1} = \varphi_i \wedge \varphi_j$ for $i,j \leq n$, the theorem obviously holds.

If $\varphi_{N+1} = M, v \models \Diamond^n_{l_{(i_2\cdots i_l)}} \varphi_i$ for $i \leq n$. We have:

$$M, v \models \Diamond^n_{l_{(i_2\cdots i_l)}} \varphi \iff \bigvee_{0 \leq i \leq n} (M, v \models \Diamond^i_{l_{(i_2\cdots i_l,0)}} \vee M, v \models \Diamond^{n-i}_{l_{(i_2\cdots i_l,1)}}) \tag{36}$$

Hence the theorem holds when for every formula when $N \leq n + 1$, therefore If $l \leq 2$, every FO formula that can be expressed by (l+1)-div graded modal logic classifier can also be expressed by l-div graded modal logic classifier. $\square$

**Corollary D.1.** *If $l \geq 2$, every formula FO that can be expressed by graded modal logic is able to be expressed by l-div graded modal logic.*

*Proof.* Notice if $l \geq 2$, every formula FO that can be expressed by 2-div graded modal logic is able to be expressed by l-div graded modal logic and:

$$M, v \models \Diamond^n \varphi \iff \bigvee_{0 \leq i \leq n} (M, v \models \Diamond^i_{l_{(0)}} \vee M, v \models \Diamond^{n-i}_{l_{(1)}}) \tag{37}$$

Hence every formula FO that can be expressed by graded modal logic is able to be expressed by 2-div graded modal logic. $\square$

## D.3 PROOFS OF NEIGHBOR L-DIVISION COLOR REFINEMENT ALGORITHM

**Lemma D.2.** *For $l \geq 2$, there exists a pair of non-isomorphic graphs that l-division color refinement algorithm outputs "possibly isomorphic" while (l+1)-division color refinement algorithm outputs "non-isomorphic"*

*Proof.* For $l \geq 2$, consider the following pair of graphs: l+1 circles with l+2 in length and l+2 circles with l+1 in length. $\square$

**Theorem D.3.** *For $l \geq 2$, (l+1)-division color refinement algorithm is strictly more powerful than l-division color refinement algorithm.*

*Proof.*

**Lemma D.3.** *For two node color refine algorithms $L_1$ and $L_2$, each algorithm will allot every node $v$ a color as $col(v)^{L_1}$ and $col(v)^{L_2}$, if there exists a surjection $\varphi(\cdot)$ that $\varphi(col(v)^{L_2}) = col(v)^{L_1}$, then $L_2$ is more powerful than $L_1$.*

*Proof.* Two node color refine algorithms $L_1$ and $L_2$ will decide the pair of graphs $(G, G')$ is isomorphism if there exists a surjection $\psi(\cdot)$ that $col(v) = \psi(col(v'))$ for every node $v \in G, v' \in G'$, Since node color refine algorithm is invariant, for a pair of isomorphism graphs, node color refine algorithm will always decide the pair of graphs is isomorphism. Hence if there exists a function $\varphi(\cdot)$ that $\varphi(col(v)^{L_2}) = col(v)^{L_1}$ for node in both $(G, G')$ and algorithm $L_2$ decide the pair of graphs is isomorphism, then there exists a surjection $\psi_2(\cdot)$ that $col^{L_2}(v) = \psi_2(col^{L_2}(v'))$ for every node $v \in G, v' \in G'$, hence we can define $\psi_1(\cdot) = \psi_2(\varphi(\cdot))$ , then $\psi_2(\cdot)$ that $col^{L_2}(v) = \psi_1(col^{L_2}(v'))$ for every node $v \in G, v' \in G'$, and algorithm will $L_1$ decide the pair of graphs is isomorphism, so $L_2$ is more powerful than $L_1$. $\square$

Recall of the iteration of l-div algorithm:

$$c_v^t \leftarrow hash(c_v^{t-1}, \{\{c_w^{t-1} : w \in \mathcal{N}_{l_{(i_2, i_3 \cdots i_l)}}(v)\}\}), \forall v \in V, (i_2, i_3 \cdots i_l) \in (0, 1)^{l-1} \tag{38}$$

for every t, construct a multisets surjection $\varphi_{l+1}^t$ as:

$$\varphi_{l+1}^t(c_v^{t-1}, \{\{c_w^{t-1} : w \in \mathcal{N}_{l+1_{(i_2, i_3 \cdots i_l, i_{l+1})}}(v)\}\}) = (c_v^{t-1}, \{\{c_w^{t-1} : w \in \mathcal{N}_{l+1_{(i_2, i_3 \cdots i_l, 0)}}(v) \quad or \quad w \in \mathcal{N}_{l+1_{(i_2, i_3 \cdots i_l, 1)}}(v)\}\})$$

since $hash()$ is an injective, hence there exists $\varphi_{l+1}'^t(c_{l+1}(v)^t) = c_l(v)^t$. Denote the output color from (l+1)-division color refinement algorithm and l+-division color refinement algorithm as $c_{l+1}(v)$ and $c_l(v)$, then there exists $\varphi_{l+1}'(c_{l+1}(v)) = c_l(v)$, by lemma C.2 , (l+1)-division color refinement algorithm is strictly more powerful than l-division color refinement algorithm $\square$

## D.4 PROOFS OF NEIGHBOR DIVISION FRAMEWORK

**Theorem D.4.** *A logical classifier is captured by l-Div-AC-GNNs if and only if it can be expressed in l-div graded modal logic.*

*Proof.* $\longrightarrow$ We wiil proof if a FO formula can be expressed l-div-graded modal logic, then there exists a l-div-AC-GNN to satisfy.

Given a l-graded modal logic with formula as $\gamma(x)$, we can decompose $\gamma(x)$ as a squence of $\gamma_i()$, denote as $sub(\gamma) = (\gamma_1, \gamma_2, \cdots \gamma_T)$ and $\gamma = \gamma_T$, every $\gamma_k$ is formed as the the following one:$color(x)$

, $\neg\gamma_i$ ,$\gamma_i \wedge \gamma_j$,$\neg\gamma$, $\gamma_i(x) \vee \gamma_j$ and $\Diamond^{\geq N}_{l_{(i_2,i_3,\cdots i_l)}}\gamma_i$ $(i, j \leq k)$, where if $\gamma_i()$ is subformula of $\gamma_j()$ then $i \leq j$.

We will construct a T-layers l-div-AC-GNN Model with feature vector $X \in (0,1)^{V \times T}$ and $X^i_{(v,t)} = 1(X^i_{(v,t)}$ is node v's $t^t h$ component of feature x at layer i) means $v| = \gamma_i$ otherwise $X^{(v,T)=0}$. Since $\gamma = \gamma_T$, we will only need to make sure $X_{(T,v)} = 1$ if and only if $v| = \gamma$. Hence, for each layer i, we define a l-div-AC-GNN's with aggregation and combine functions as follow:

$$m_v^{(i,((i_2,i_3\cdots,i_l))} = AGGREGATE^{(i,((i_2,i_3\cdots,i_l))}(X_v^i) = \sum_{u \in N_{(i_2,i_3\cdots,i_l)}(v)} (X_u^i) \quad (40)$$

$$COMBINE(X_v^i, \{m^{(i,((i_2,i_3\cdots,i_l))}\}) = \sigma(X_v^i A + \sum_{(i_2,i_3\cdots,i_l)\in(0,1)^{l-1}} m_v^{(i,((i_2,i_3\cdots,i_l))}B^{(i_2,i_3\cdots,i_l)}+c) \quad (41)$$

$A, B \in R^{T \times T}$ are learnable matrix and $c \in R^T$ is learnable vector, Let RELU activation function $\sigma(x) = min(max(0, x), 1)$. The parameter of A,B,c are defiined as followed:

if $\gamma_i = Color(x)$, then let $A_{(i,i)} = 1$

if $\gamma_i = \gamma_j \wedge \gamma_k$, then let $A_{(j,i)} = 1$, $A_{(k,i)} = 1$ and $c_i = -1$

if $\gamma_i = \neg\gamma_j$, then let $A_{(j,i)} = -1$ and $c_i = 1$

if $\gamma_i(x) = \Diamond^{\geq N}_{l_{(i_2,i_3,\cdots i_l)}}\gamma_i$, then let $B^{(i_2,i_3\cdots,i_l)}_{(j,i)} = 1$ and $c_i = N - 1$

and the rest of value not mentioned in the $i^t h$ columns of A,B,c are 0.

We now prove that model $M_{l-div}$ is able to capture logic $\gamma$. Given a colored directed graph $\vec{G}$, set initial feature $X_v^0 = (X_{(v,1)}^0, X_{(v,2)}^0 \cdots X_{(v,T)}^0)$ that $X_{(v,t)}^0 = 1$ iff $\gamma_i$ is fundamental logic (as $color(x)$ ) and $v| = \gamma_i$ else $X_{(v,t)}^0 = 0$. After T rounds iteration, as equation (41) and (42), we will prove equation (43) holds:

$$when \quad t \in \{1, 2\cdots, i\}, X_{(v,t)}^i = 1 \quad if \quad v| = \gamma_i otherwise X_v^i = 0 \quad (42)$$

That means after at least t rounds iteration, l-div-AC-GNN is able to capture $\gamma_t$. By equation (41) and (42), the iteration expression of $X_{(v,t)}^i$ is

$$X_{(v,t)}^i = \sigma(X_v^i A + \sum_{(i_2,i_3\cdots,i_l)\in(0,1)^{l-1}} \sum_{u \in N_{(i_2,i_3\cdots,i_l)}(v)} (X_u^i)B^{(i_2,i_3\cdots,i_l)} + c) \quad (43)$$

When t=1,$\gamma_1$ has one subformula as $\gamma_1$ is $color()$, if $\gamma_i(x) = Color(x)$, then $A^{(1,1)} = 1$ and the rest parameter of $1^{th}$ columns equals to 0 as $A^{(i,1)} = 0(i \leq 2)$, $B^{(i,1)} = 0$ and $c^{(1)} = 0$, equation (44) can be rewritten as

$$X_{(v,1)}^1 = \sigma(X_{(v,1)}^0) \quad (44)$$

Given initial feature $X_{(v,1)}^0 = 1$ if $color(v)$ is true and $X_{(v,1)}^0 = 0$ otherwise, hence equation (43) holds when t=1. Assume equation (43) holds when t=k, we now prove it holds when t=k+1, $\gamma_{k+1}$ can be expressed by $(\gamma_1, \gamma_2, \cdots \gamma_k)$ as follow

Case 1: if $\gamma_{k+1}(x) = Color(x)$, it is same argument as $\gamma_0(x)$, hence equation (43) holds.

Case 2:if $\gamma_{k+1}(x) = \gamma_i \wedge \gamma_j(i, j \leq k + 1)$, then $A_{(i,k+1)} = 1$, $A_{(j,k+1)} = 1$ and $c_{k+1} = -1$,also $A_{(t,k+1)} = 0$ for every $1 \leq t \leq T, i, j \neq t$ and $B^{(i_2,i_3\cdots,i_l)}_{r,k+1} = 0$ for every $1 \leq r \leq T$ and $(i_2, i_3 \cdots, i_l) \in (0,1)^{l-1}$. $X_{(v,k+1)}^{k+1}$ can be expressed as

$$X_{(v,k+1)}^{k+1} = \sigma(X_{(v,i)}^k + X_{(v,j)}^k - 1) \quad (45)$$

By the assumption equation (43) holds when t=k, then $X_{(v,i)}^k = 1$ and $X_{(v,j)}^k = 1$ iff $v| = \gamma_i$ and $v| = \gamma_i$. Since RELU activation function is $\sigma(x) = min(max(0, x), 1)$, if $X_{(v,k+1)}^{k+1} = 1$, then we

can deduce $X^k_{(v,i)} + X^k_{(v,j)} - 1 \geq 1$, hence $X^k_{(v,i)} = 1$ and $X^k_{(v,j)} = 1$ while indicating $v| = \gamma_i$ and $v| = \gamma_i$, then $v| = \gamma_{k+1}$. On the other hand, if $X^{k+1}_{(v,k+1)} = 0$, we can deduce $X^k_{(v,i)} + X^k_{(v,j)} - 1 < 1$, hence either $X^k_{(v,i)} = 0$ or $X^k_{(v,j)} = 0$, indicating $v| \neq \gamma_i$ or $v| \neq \gamma_i$, hence $v| \neq \gamma_{k+1}$. Now we have proved equation (43) holds when $\gamma_{k+1}(x) = \gamma_i \wedge \gamma_j (i, j \leq k+1)$.

Case 3: if $\gamma_{k+1}(x) = \neg\gamma_i (i, j \leq k+1)$, then $A_{(j,i)} = -1$ and $c_i = 1$. $X^{k+1}_{(v,k+1)}$ can be expressed as

$$X^{k+1}_{(v,k+1)} = \sigma(-X^k_{(v,i)} + 1) \tag{46}$$

By the assumption equation (43) holds when t=k, then $X^k_{(v,i)} = 0$ iff $v| \neq \gamma_i$. Since RELU activation function is $\sigma(x) = min(max(0,x),1)$, if $X^{k+1}_{(v,k+1)} = 1$, then we can deduce $-X^k_{(v,i)} + 1 \geq 1$, hence $X^k_{(v,i)} = 0$ while indicating $v| \neq \gamma_i$, then $v| = \gamma_{k+1}$. On the other hand, if $X^{k+1}_{(v,k+1)} = 1$, we can deduce $-X^k_{(v,i)} + 1 < 1$, hence $X^k_{(v,i)} = 1$, indicating $v| = \gamma_i$, hence $v| \neq \gamma_{k+1}$. Now we have proved equation (43) holds when $\gamma_{k+1}(x) = \neg\gamma_i (i, j \leq k+1)$.

Case 4: if $\gamma_{k+1}(x) = \Diamond^{\geq N}_{l_{(i_2,i_3,\cdots i_l)}} \gamma_i$, then $B^{(i_2,i_3\cdots,i_l)}_{(i,k+1)} = 1$ and $c_i = N - 1$, also $A_{(t,k+1)} = 0$ for every $1 \leq t \leq T$ and $B^{(i'_2,i'_3\cdots,i'_l)}_{r,k+1} = 0$ for every $1 \leq r \leq T$, $(i'_2, i'_3 \cdots, i'_l) \in (0,1)^{l-1} while (i'_2, i'_3 \cdots, i'_l) \neq (i_2, i_3 \cdots, i_l)$ and $B^{(i_2,i_3\cdots,i_l)}_{(r,k+1)} = 0$ for every $1 \leq r \leq T$, $r \neq i$. $X^{k+1}_{(v,k+1)}$ can be expressed as:

$$X^{k+1}_{(v,k+1)} = \sigma(\sum_{u \in N_{(i_2,i_3\cdots,i_l)}(v)} X^i_u - N + 1)) \tag{47}$$

By the assumption equation 43 holds when t=k, then $X^i_u = 1$ iff $u| = \gamma_i$. Since RELU activation function is $\sigma(x) = min(max(0,x),1)$, if $X^{k+1}_{(v,k+1)} = 1$, then we can deduce $\sum_{u \in N_{(i_2,i_3\cdots,i_l)}(v)} X^i_u - N + 1) \geq 1$, hence the number of node in set $\{u \mid u \in \mathcal{N}_{L_{i_1,i_2\cdots i_t}}(v), u| = \sigma_i\}$ is at least N, by **??** $v| = \gamma_{k+1}$. On the other hand, if $X^{k+1}_{(v,k+1)} = 0$, we can deduce $\sum_{u \in N_{(i_2,i_3\cdots,i_l)}(v)} X^i_u - N + 1) < 1$, hence the number of node in set $\{u \mid u \in \mathcal{N}_{L_{i_1,i_2\cdots i_t}}(v), u| = \sigma_i\}$ is at most N-1, by **??** $v| \neq \gamma_{k+1}$. Now we have proved equation 43 holds when $\gamma_{k+1}(x) = \exists^{\geq N}(E_{(i_2,i_3,\cdots i_l)}(x,y) \wedge \gamma_i(y))$.

Now we have proved equation 43 when t=k+1, hence at $T^{th}$ layer, we have $X^T_{(v,T)} = 1$ if $v| = \gamma_T$ otherwise $X^i_v = 0$, that means can capture l-div graded modal logic classifier $\gamma$, since $\gamma$ is arbitrary, then every l-div graded modal logic classifier can be captured by a l-Div-AC-GNN.

$\Longleftarrow$: We only need to prove that if a logical classifier is not able to be expressed by l-div graded modal logic then it cannot be captured by l-Div-AC-GNN.

**Theorem D.5.** *Given simple, directed and node-colored graph $\vec{G}$ and $\vec{G'}$, then for any l-div AC-GNN with that maps two graph $\vec{G}$ and $\vec{G'}$ to same features if l-neighbor division color refinement algorithm decides $\vec{G}$ and $\vec{G'}$ are isomorphism.*

*Proof.* We will show that if node $u$ and $u'$ in $\vec{G}$ and $\vec{G'}$ gets same labels at t iteration $c^t_u = c^t_{u'}$, for any l-div AC-GNN will always obtains same features for node $u$ and $u'$ $h^t_u = h^t u'$ at t layer. Suppose at t layer, there exists node $u$ and $u'$ l-div AC-GNN obtains different features $h^t_u = h^t_{u'}$ and AC-GNN gets same features from layer $t-1$ to 0, but l-neighbor division color refinement algorithm assigns node $u$ and $u'$ the same label at t layer. Since different multisets get different new labels for every iteration, at t iteration in l-neighbor division color refinement algorithm, node $u$ and $u'$ gets same labels $c^t_u \leq= c^t u'$, indicating that

$$(c^{t-1}_u, \{\{c^{t-1}_v : v \in N_{(i_2,i_3\cdots i_l)}(u)\}, (i_2, i_3 \cdots i_l) \in (0,1)^{l-1}\}) = (c^{t-1}_{u'}, \{\{c^{t-1}_v : v \in N_{(i_2,i_3\cdots i_l)}(u')\}, (i_2, i_3 \cdots i_l) \in (0,1)^{l-1}\}) \tag{48}$$

On the other hand, at layer t in l-div AC-GNN

$(h_u^{t-1}, \{\{h_v^{t-1} : v \in N_{(i_2,i_3\cdots i_l)}(u)\}, (i_2, i_3 \cdots i_l) \in (0,1)^{l-1}\}) = (h_{u'}^{t-1}, \{\{h_v^{t-1} : v \in N_{(i_2,i_3\cdots i_l)}(u')\}, (i_2, i_3 \cdots i_l) \in (0,1)^{l-1}\})$ (49)

By the assumption, we have $h_v^{t-1} = h_{v'}^{t-1}$ for every node in $\vec{G}$ and $\vec{G}'$, since the same neighborhood features, neighborhood and GNN are applied to generates the same features $h_u^t = h_{u'}^t$ at $t^{th}$ layer. In this way, We prove that l-div AC-GNN obtains same features for node $u$ and $u'$ if them gets same labels at $t$ iteration in l-neighbor division color refinement algorithm. This means there exists a mapping $h_v^t = \psi(c_v^t)$ for node $v$ in $\vec{G}$ and $\vec{G}'$. Therefore, we have

$(h_u^t, \{\{h_v^t : v \in N_{(i_2,i_3\cdots i_l)}(u)\}, (i_2, i_3 \cdots i_l) \in (0,1)^{l-1}\}) = (\psi(c_u^{t-1}), \{\{\psi(c_v^t) : v \in N_{(i_2,i_3\cdots i_l)}(u')\}, (i_2, i_3 \cdots i_l) \in (0,1)$
(50)

Therefore, if l-neighbor division color refinement algorithm decides $\vec{G}$ and $\vec{G}'$ are isomorphism, then for every iteration node $u$ and $u'$ in $\vec{G}$ and $\vec{G}'$ gets same labels, the output of $\vec{G}$ and $\vec{G}'$ in l-div AC-GNN will get same features.

$\square$

To prove this theorem, we have the definition as follow:

**Definition D.1.** (*l-div g-bisimulations*) *We define l-div graded logic model as* $M = (W, E_{l_{(i_2,i_3\cdots i_l)}}, V)$ *where* $W$ *is a non-empty set of states,* $E_{(i_2,i_3\cdots i_l)}$ *is a binary relation set on* $W$ *write as* $xE_{(i_2,i_3\cdots i_l)}y$, $xE_{(i_2,i_3\cdots i_l)}^\bullet Y$ *denote* $xE_{(i_2,i_3\cdots i_l)}y$ *for all* $y \in Y$ *if* $xE_{(i_2,i_3\cdots i_l)}y$ *is true then* $\neg xE_{(i_2',i_3'\cdots i_l')}y$ *is true iff* $(i_2, i_3 \cdots i_l) \neq (i_2', i_3' \cdots i_l')$ *and valuation* $V$ *is a function assigning a subset of* $W$ *to every proposition letter. Based on g-bisimulations, we propose l-div g-bisimulations: Given two models* $M = (W, E, V)$ *and* $M' = (W', E', V')$, *l-div g-bisimulations between* $M$ *and* $M'$ *is a tuple of* $Z = (Z_1, Z_2, \cdots)$ *relation* $Z_n \subseteq W \times W'$ *satisfying the following requirements:*

*1. Z is non-empty;*

*2. if* $xZ_nx'$, *then* $x| = p$ *iff* $x'| = p$, *for all proposition letters p;*

*3. if* $xZ_nx'$ *and* $xE_{(i_2,i_3\cdots i_l)}^\bullet Y$, *then there exists* $Y' \in W'$ *with* $YZ_nY'$ *and* $x'E_{(i_2,i_3\cdots i_l)}^\bullet Y'$

*4. if* $xZ_nx'$ *and* $x'E_{(i_2,i_3\cdots i_l)}^\bullet Y'$, *then there exists* $Y \in W$ *with* $YZ_nY'$ *and* $xE_{(i_2,xZ_ix'i_3\cdots i_l)}^\bullet Y$

*5. if* $xZ_nx'$ *then for every* $1 \leq m \leq n$

*(a) for every* $x \in X$ *there exists* $x' \in X'$ *with* $\{x\}Z_n\{x'\}$, *and*

*(b) for every* $x' \in X'$ *there exists* $x \in X$ *with* $\{x\}Z_n\{x'\}$.

If there is a l-div g-bisimulation Z between M and M0 as $wZ$, we denote as $M, \omega \rightleftharpoons_g^{l-div} M', \omega'$. The l-div graded modal type of a state is the set of all graded modal formulas it satisfies: $tp_l(\omega) = \{\varphi \in L_{GML}, \omega| = \varphi\}$; if necessary we record the model M in which $\omega$ lives as a subscript: $tp_l(\omega)$. Two states $\omega$, v are l-div graded modally equivalent if $tp_l(\omega) = tp_l(v)$ denote as $\omega \equiv_g^{l-div} v$.

**Lemma D.4.** *Let* $M$ *and* $M'$ *be two* $\omega$-*saturated models, and let* $\omega \in W$, $\omega' \in W'$. *Then* $M, \omega \rightleftharpoons_g^{l-div} M', \omega'$ *iff* $\omega \equiv_g^{l-div} \omega'$

*Proof.* $\Rightarrow$: Assume it holds for $degree(\varphi) \leq k$ if there exists a l-div g-bisimulations between $M$ and $M'$, to prove it holds for $degree(\varphi) \leq k+1$, it is obvious that if $\varphi, \psi \in tp_l(\omega)$ then $\neg\varphi, \varphi\wedge\psi \in tp_l(\omega)$ hence the atomic and boolean cases are trivial. For modal case, if $\omega| = \exists^{\geq n}v(\varphi\wedge E_{(i_2,i_3\cdots i_l)(\omega,v)})$, there exists $|Y| \in 2^W$ $xE_{(i_2,i_3\cdots i_l)}^\bullet V$ and $V| = \varphi$, since $M, \omega \rightleftharpoons_g^{l-div} M', \omega'$, by the definition of l-div g-bisimulations, there exists $|Y| \in 2^W$ for $\omega'E_{(i_2,i_3\cdots i_l)}^\bullet V'$ and $VZ_nV'$, as $VZ_nV'$ hence $V'| = \varphi$ for $degree(\varphi) \leq k$, then $\omega'| = \exists^{\geq n}v'(\varphi \wedge E_{(i_2,i_3\cdots i_l)(\omega',v')})$ for $\omega'$, hence

it holds for $degree(\varphi) \leq k+1$, as $tp_l(\omega) = \bigcup_{1 \leq k} \{\varphi \in L_{GML}, \omega| = \varphi, degree(\varphi) = k\}$, this implies $M, \omega \rightleftharpoons_g^{l-div} M', \omega' \Rightarrow \omega \equiv_g^{l-div} v$.

$\Leftarrow$:assume that for every finite set W and W', define the tuple of bijection relation $Z = (Z_0, Z_1, Z_2, \cdots)$ as follow:

(1)$\{\omega\}Z_0\{\omega'\}$

(2) $vZ_n v'$ iff $vZ_{n-1}v'$ or $\{t\}Z_{i-1}\{t'\}$ $E_{(i_2,i_3\cdots i_l)}(t,v), E_{(i_2,i_3\cdots i_l)}(v',t')$ and $tp_l(v) = tp_l(v')$

First we will prove if $\{t\}Z_{i-1}\{t'\}$ $E_{(i_2,i_3\cdots i_l)}(t,v)$ there exists $v'$ that $E_{(i_2,i_3\cdots i_l)}(v',t')$ and $tp_l(v) = tp_l(v')$. Assume $\omega E_{(i_2,i_3\cdots i_l)}^{\bullet} V^{(i_2,i_3\cdots i_l)}$ and $|V| = n^{(i_2,i_3\cdots i_l)}$, define the set of every type $tp(V^{(i_2,i_3\cdots i_l)})$ as:

$$\{T_1^{(i_2,i_3\cdots i_l)}, \cdots T_s^{(i_2,i_3\cdots i_l)}\} = \{tp(v)|v \in V^{(i_2,i_3\cdots i_l)}\} \quad T_i^{(i_2,i_3\cdots i_l)} \neq T_j^{(i_2,i_3\cdots i_l)} \; if \; i \neq j \quad (51)$$

and number of every type $n_i^{(i_2,i_3\cdots i_l)}$:

$$n_i = |\{v \in V | tp(v) = T_i^{(i_2,i_3\cdots i_l)}\}| \quad (52)$$

If $E_{(i_2,i_3\cdots i_l)}(\omega,v)$, then there exists $i$ that $tp(v) = T_i^{(i_2,i_3\cdots i_l)}$, since $\omega \equiv_g^{l-div} v$, any $\varphi \in tp(v)$ we have $\omega| = \exists^{\geq n_i} v(\varphi \wedge E_{(i_2,i_3\cdots i_l)}(\omega,v))$, hence $\omega'| = \exists^{\geq n_i} v'(\varphi \wedge E_{(i_2,i_3\cdots i_l)}(\omega',v'))$, so when i=1 there exists a set of $\omega'$'s $E_{(i_2,i_3\cdots i_l)}$ neighbor $v \in V'$ that $E_{(i_2,i_3\cdots i_l)}(v',t')$ and $tp_l(v) = tp_l(v')$. Assume it holds when i=k, then $tp_l(t) = tp_l(t')$ same argument as i=1, $t| = \exists^{\geq n_i} v(\varphi \wedge E_{(i_2,i_3\cdots i_l)}(t,v))$ implies $t'| = \exists^{\geq n_i} v'(\varphi \wedge E_{(i_2,i_3\cdots i_l)}(t',v'))$, there exists $v'$ that $E_{(i_2,i_3\cdots i_l)}(v',t')$ and $tp_l(v) = tp_l(v')$. Notice that we can swap t and t', so there exists a bijection relation between E(t) and E(t') for $(t,t') \in Z_{n-1}$, we donote the bijection relation as $Z_{n-1}^{neighbor}$. Notice that $Z_n = Z_{n-1} \bigcup Z_{n-1}^{neighbor}$, hence for every $\omega \equiv_g^{l-div} \omega'$ we can construct the bijection relation Z in between M and M'.

Since W and W' are based on connected graph, there exists n that $Z_n = W \times W'$ We now prove that $Z_n$ is l-div g-bisimulations, obvious item 1 is qualified, for item 2 there exists certain i for $vZ_n v'$ that $(v,v') \notin Z_{i-1}$ and $(v,v') \in Z_i$ then $tp_l(v) = tp_l(v')$ hence $x| = p$ iff $x'| = p$.

For item 3,4 if $\{v\}Z_n\{v'\}$, there exists certain i that $(v,v') \notin Z_{i-1}$ and $(v,v') \in Z_i$, if $vE^{\bullet}Q$, since $Z_i$ is bijection, for every $q \in E_{(i_2,i_3\cdots i_l)}(Q)$ there is a unique $q'$ that $\{q\}Z_i\{q'\}$, by the defintion $E_{(i_2,i_3\cdots i_l)}(v',q')$ and $tp_l(q) = tp_l(q')$, denote the set of $q'$ as $Q'$, hence $v'E^{\bullet}Q'$ and $QZ_iQ'$, notice $Z_i \subseteq Z_n$, hence $QZ_nQ'$. Hence $Z_n$ satisfies item 3,4.

For item 5, it obviously holds for $Z_0$, if $XZ_nX'$ then for every $x \in X$ there exists certain i that $X_{i-1}Z_{i-1}X'_{i-1}$ $x \in X_{i-1}$ and $X_iZ_iX'_i$ $x \notin X_i$, then there exists a unique x' that $X_{i-1}Z_{i-1}X'_{i-1}$ $x' \in X'_{i-1}$. Swapping the position of x and x' will imply b. Hence $Z_n$ satisfies item 5. Now on we have proved $Z_n$ is l-div g-bisimulations. $\square$

**Proposition 6.** *The Compactness Theorem (Malcev)Let be countable, and let D be a countably incomplete ultrafilter over a set I. Then for every family $\Psi$, $i \in I$, of models for , the ultraproduct $\prod_D i$ is $\omega$-saturated.*

**Theorem D.6.** *(Invariance). Assume that L1 is countable. An L1-formula $\alpha(x)$ is (equivalent to the translation of ) a l-div graded modal formula iff it is invariant under l-div g-bisimulations.*

*Proof.* $\Rightarrow$ has been proved in lemma D.4. For the $\Leftarrow$, if $\alpha(\omega)$ is invariant under l-div g-bisimulations, construct the set of l-div graded formula consequence of $\alpha(\omega)$ as follow:

$$\Phi_{l-g}(\alpha) = \{\varphi|\varphi(\omega)| = \alpha(\omega), \varphi \in L_{l-div\, g}\} \quad (53)$$

Notice that $\alpha(\omega)$ is l-div graded formula iff $\alpha(\omega) \in \Phi_{l-g}(\alpha)$, specifically, for any model M and $\omega$, formula$M| = \Phi_{l-g}(\alpha)[\omega] \Rightarrow M| = \alpha(\omega)$ implies $\alpha(\omega)$ is l-div graded formula.

If for any model M and $\omega$, $M| = \Phi_{l-g}(\alpha)[\omega] \Rightarrow M, \omega| = \neg\alpha$, then $\neg\alpha(\omega) \in \Phi_{l-g}(\alpha) \Rightarrow \alpha(\omega) \in \Phi_{l-g}(\alpha)$. Hence if $\alpha(\omega)$ is not l-div graded formula, for any $\omega$, $M| = \Phi_{l-g}(\alpha)[\omega]$, there exists

$N, v, N| = \Phi_{l-g}(\alpha)[v]$ and $N, v| = \alpha$, since M,N is $\omega-$saturated [proposition6], consider the $\omega-$saturated extension of $(N^+, v)$ and $(M^+, \omega)$, then $tp_{M^+}^{l-div}(\omega) = tp_{N^+}^{l-div}(v) \Rightarrow M^+, \omega \rightleftharpoons_g^{l-div} N^+, v$. Since M and N is $\omega-$saturated models, $\{\alpha(\omega)\}$ is finite and $N, v| = \alpha$. Hence we have $N^+, v| = \alpha$, which implies $M^+| = \alpha(\omega)$ then $M| = \alpha(\omega)$, hence $\alpha(\omega) \in \Phi_{l-g}(\alpha)$ ,$\alpha(\omega)$ is l-div graded formula.

$\square$

**Theorem D.7.** *Let M and M' be graph $G_0$'s and $G_1$'s l-div graded models, then l-division color refinement algorithm decides $G_0$ and $G_1$ are isomorphism if and only if M and M' are under l-div g-bisimulations.*

*Proof.* $\Longleftarrow$ If M and M' are under l-div g-bisimulations, then there exists a bijection $v_0 Z v_1$ between node set $V_0$ and $V_1$ in $G_0$ and $G_1$. We will prove if $v_0 Z v_1$ then the output color of $v_0$ and $v_1$ are the same. By item 2 in2.4, the initial of $v_0$ and $v_1$ are the same. Assume at layer t, the color of $v_0$ and $v_1$ are the same, then by item 3 and 4 in2.4,$(c_{v_0}^{t-1}, \{\{c_{w_0}^{t-1} : w \in \mathcal{N}_{l_{(i_2,i_3\cdots i_l)}}(v_0)\}\})$ and $(c_{v_1}^{t-1}, \{\{c_{w_1}^{t-1} : w \in \mathcal{N}_{l_{(i_2,i_3\cdots i_l)}}(v_1)\}\})$ are the same. Hence, it holds at layer t+1. Then the output color of $v_0$ and $v_1$ are the same.l-division color refinement algorithm decide $G_0$ and $G_1$ are isomorphism $\Longrightarrow$ If l-division color refinement algorithm decides $G_0$ and $G_1$ are isomorphism, then there exists a bijection $v_0 Z v_1$, we will prove it is l-div g-bisimulations.

For item 1, it is obviously non-empty

For item 2, since the color of $v_0$ and $v_1$ are the same, which indicating that their initial of $v_0$ and $v_1$ are the same. Hence $v_0| = p$ iff $v_1| = p$.

For item 3 and 4, if $x Z_n x'$ and $x E_{(i_2,i_3\cdots i_l)}^{\bullet} Y$ and there does not exist $Y' \in W'$ with $Y Z_n Y'$ and $x' E_{(i_2,i_3\cdots i_l)}^{\bullet} Y'$. Then recall the equation of iteration in l-division color refinement algorithm:

$$c_v^t \leftarrow hash(c_v^{t-1}, \{\{c_w^{t-1} : w \in \mathcal{N}_{l_{(i_2,i_3\cdots i_l)}}(v)\}\}), \forall v \in V, (i_2, i_3\cdots i_l) \in (0,1)^{l-1} \quad (54)$$

Then the color of $v_0$ and $v_1$ will not be same, hence item 3 and 4 holds.

For item 5, for every $v_0 \in V_0$, Z will maps $V_0$ to $V_1$ and $v_0$ to $v_1$, since Z is a bijection, then $v_1 \in V_1$, hence item 5 holds $\square$

We now prove if a formula can be captured by a l-div GNN then it can be expressed by l-div graded modal logic: assume there is a formula $\alpha$ can be captured by a l-div GNN but cannot be expressed by l-div graded modal logic then by theorem D.6 $\alpha$ there is a pair of graphs $G_0$ and $G_1$ and node $v_0 \in G_0$ and $v_1 \in G_1$, then there is a l-div g-bisimula between their induced models M and M' but $M, v_0| = \alpha$ ,$M', v_1| \neq \alpha$. By theorem D.7, l-division color refinement algorithm decides $G_0$ and $G_1$ are isomorphism . And by theorem D.5, the outputs of any l-div GNN for $G_0$ and $G_1$ will always be the same which means l-div GNN is not able to capture $\alpha$.Therefore we have proved both directions of the theorem, hence a logical classifier is captured by l-Div-AC-GNNs if and only if it can be expressed in l-div graded modal logic.

$\square$

**Theorem D.8.** *Given a AC-GNN of countable additivity, then its l-div framework inherits three Properties: invariance and equivariance, approximate and logic expressive power .*

*Proof.* Given an AC-GNN of countable additivity with T layers whose aggregate and combine function is as follow, the feature of node $v$ at layer t is denoted as $x^t(v)$:

$$x^t(v) = COMB(x^{t-1}(v), AGGRE(\{\{x^{t-1}(\omega), \omega \in N(v)\}\})) \quad 1 \geq t \geq T \quad (55)$$

Denote the feature of node $v$ at layer t in AC-GNN's l-div framework as $x'^t(v)$, we will prove its l-div framework inherits three Properties: invariance and equivariance, approximate and logic expressive power:

$$m_{l_{(i_2\cdots i_l)}}^t(v) = AGGRE^t(\{\{x'^{t-1}(\omega), \omega \in N_{l_{(i_2\cdots i_l)}}(v)\}\})) \quad 1 \geq t \geq T \quad (56)$$

$$x'^t(v) = COMB^t(x^{t-1}(v), \sum_{(i_2 \cdots i_l) \in (0,1)^{l-1}} m_{l_{(i_2 \cdots i_l)}}^t(v))) \quad 1 \geq t \geq T \tag{57}$$

**Invariance and Equivariance:** Given an AC-GNN, we will prove its the l-div framework will inherit its invariance and equivariance. In graph level task, let $S_I^{AC-GNN}(n)$ denoted AC-GNN'S invariance permutation set and final output as $AGGRE_{graph-level v \in V}(x^t(v))$. Then $\forall \sigma_I \in S_I^{AC-GNN}(n)$, for every node $v$ denote $\sigma_I(v)$ as $\sigma_I$ maps $v$ to, then :

$$AGGRE_{graph-level v \in V}(x^T(\sigma_I(v))) = AGGRE_{graph-level v \in V}(x^T(v)) \tag{58}$$

By equation 58

$$x'^t(\sigma_I(v)) = COMB^t(x^{t-1}(\sigma_I(v)), \sum_{(i_2 \cdots i_l) \in (0,1)^{l-1}} AGGRE^t(\{\{x'^{t-1}(\omega), \omega \in N_{l_{(i_2 \cdots i_l)}}(\sigma_I(v))\}\}))$$

=COMB(x$^{t-1}(\sigma_I(v)), AGGRE(\{\{x^{t-1}(\omega), \omega \in N(\sigma_I(v))\}\})) = x^t(\sigma_I(v))$(59) Therefore

$$AGGRE_{graph-level v \in V}(x'^T(\sigma_I(v))) = AGGRE_{graph-level v \in V}(x'^T(v)) \tag{60}$$

In node level task, let $S_E^{AC-GNN}(n)$ denoted AC-GNN'S equivariance permutation set . Then $\forall \sigma_E \in S_E^{AC-GNN}(n)$, then :

$$x^T(\sigma_E(v)) = \sigma_E(x^T(v)) \tag{61}$$

$$x'^t(\sigma_E(v)) = COMB^t(x^{t-1}(\sigma_E(v)), \sum_{(i_2 \cdots i_l) \in (0,1)^{l-1}} AGGRE^t(\{\{x'^{t-1}(\omega), \omega \in N_{l_{(i_2 \cdots i_l)}}(\sigma_E(v))\}\}))$$

=COMB(x$^{t-1}(\sigma_E(v)), AGGRE(\{\{x^{t-1}(\omega), \omega \in N(\sigma_I(v))\}\})) = x^t(\sigma_E(v))$(62) Therefore

$$x'^T(\sigma_E(v)) = \sigma_E(x'^T(v)) \tag{63}$$

Equation 61 and 64 imply that $\sigma_I \in S_I^{AC-GNN_{l-div}}(n)$ and $\sigma_E \in S_E^{AC-GNN_{l-div}}(n)$. Now we have proved that AC-GNN's l-div framework inherits porperty of invariance and equivariance.

**Approximate:** Since for all layer t, AC-GNN of countable additivity, then:

$$AGGRE(\{\{x'^{t-1}(\omega), \omega \in N(v)\}\})) = \sum_{(i_2 \cdots i_l) \in (0,1)^{l-1}} AGGRE(\{\{x^{t-1}(\omega), \omega \in N_{l_{(i_2 \cdots i_l)}}(v)\}\})$$

=$\sum_{(i_2 \cdots i_l) \in (0,1)^{l-1}} m_{l_{(i_2 \cdots i_l)}}^t(v)$(64)

hence $x'^t(v) = x^t(v)$ for every node at each layer, therefore l-div framework inherits porperty of Approximate.

# E    PROOF OF LOGIC EXPRESSIVE POWER OF AGGREGATORS

**Theorem E.1.** *Logical classifier $L'_{GML}$ can be captured by AC-GNNs which uses mean and max aggregator .*

Therefore

*Proof.* Let $aggre(\{\{x\}\})(aggre(x) = max(\{\{x\}\})$ or $aggre(x) = mean(\{\{x\}\})$ denoted aggregator. By definition, construct an AC-GNN which will iterate the as follows:

$$X_v^t = \sigma(X_v^{t-1}A^t + aggre_{u \in N(v)}(\{\{x_u^{t-1}\}\})B^t + c^t) \tag{65}$$

$A^t$, $B^t$ are learnable matrix and $c^t$ is learnable vector, Let RELU activation function

$\sigma(x) = min(max(0, x), 1)$. The parameter of $A^t, B^t, c^t$ are defined as followed:

| | random regular | | | erdos renyi | | |
|---|---|---|---|---|---|---|
| Method | 4000 | 5000 | 6000 | 4000 | 5000 | 6000 |
| $GCN$ | $67 \pm 0.87$ | $51.9 \pm 0.5$ | $63 \pm 1.08$ | $58.5 \pm 4.38$ | $51.2 \pm 1$ | $64.75 \pm 0.75$ |
| $2 - Div\,GCN$ | **100** | **100** | **100** | **100** | **100** | **100** |
| $GIN$ | $68.13 \pm 2$ | $48.8 \pm 2.1$ | $62.33 \pm 0.84$ | $64.62 \pm 0.62$ | $54.6 \pm 1.7$ | $64.58 \pm 0.42$ |
| $2 - Div\,GIN$ | **100** | **100** | **100** | **100** | **100** | **100** |
| TAG | $65 \pm 2$ | $49.5 \pm 3.4$ | $63.5 \pm 0.33$ | $61.12 \pm 0.99$ | $53.8 \pm 0.1$ | $65.75 \pm 0.42$ |
| $2 - Div\,TAG$ | **100** | **100** | **100** | **100** | **100** | **100** |

Table 1: Accuracy(%) of detecting triangle

| | random regular | | | erdos renyi | | |
|---|---|---|---|---|---|---|
| Method | 4000 | 5000 | 6000 | 4000 | 5000 | 6000 |
| $GCN$ | $66.38 \pm 1.13$ | $49.3 \pm 0.1$ | $36.25 \pm 1$ | $59.88 \pm 1.74$ | $47.5 \pm 0$ | $37.08 \pm 0.5$ |
| $2 - Div\,GCN$ | **99.38 ± 0.63** | **96.7 ± 0.1** | **94 ± 0.83** | **95 ± 0.37** | **91.4 ± 2.4** | **80.92 ± 2** |
| $GIN$ | $66.88 \pm 2.49$ | $48.9 \pm 0.6$ | $34.58 \pm 2.84$ | $60.25 \pm 2.25$ | $46.5 \pm 0.9$ | $35.42 \pm 2.75$ |
| $2 - Div\,GIN$ | **99.88 ± 0** | **99.8 ± 0.2** | **99.33 ± 0.25** | **99.25 ± 0.37** | **98.8 ± 0.2** | **98.75 ± 0.42** |
| $TAG$ | $63.75 \pm 4$ | $49.5 \pm 2.3$ | $36.17 \pm 2.9$ | $60.88 \pm 4$ | $48.6 \pm 2.1$ | $38.5 \pm 4.3$ |
| $2 - Div\,TAG$ | **99.62 ± 0** | $98.6 \pm 0.5$ | $97.83 \pm 1.16$ | **96.25 ± 1.13** | **94.8 ± 4.6** | **84.67 ± 2.75** |

Table 2: Accuracy(%) of counting number of triangles

if $\gamma_i(x) = Color(x)$, then let $A_{(i,i)} = 1$

if $\gamma_i(x) = \gamma_j \wedge \gamma_k$, then let $A_{(j,i)} = 1$, $A_{(k,i)} = 1$ and $c_i = -1$

if $\gamma_i(x) = \gamma_j \vee \gamma_k$, then let $A_{(j,i)} = 1$ and $A_{(k,i)} = 1$

if $\gamma_i(x) = \Diamond' \gamma_j$, then let $B^t_{(j,i)} = 1$ and $c_i = N - 1$

and all the whule values in the 't-th iteration of $A^t, B^t, c^t$ are 0. Then it is easy to prove AC-GNN

$\square$

**Logic Expressive Power:** $\forall \alpha \in tp(AC - GNN)$, $\alpha(v)$ is $true$ iff $x^T(v) = 1$ and $\alpha(v)$ is $false$ iff $x^T(v) = 0$, by property Approximate, we have $x^T(v) = x'^T(v)$, therefore $\forall \alpha \in tp(AC - GNN)$, $\alpha \in tp(AC - GNN_{l-div})$, implying that $tp(AC - GNN) \subset tp(AC - GNN_{l-div})$ $\square$

# F  DETAILS FOR EXPERIMENTAL SETTING AND RESULTS

We utilize synthetic data to perform experiment to validate our result. We perform the experiment using common model GIN,GCN,GAT and their 2-division version. Our experiments were implemented in the PyTorch Geometric library (Fey & Lenssen, 2019) and DEEP GRAPH LIBRARY ( Minjie & Gan 2021) and utilize networkx to generate synthetic graphs, severally with 4000,5000,6000 nodes with 0.006,0.005,0.004. Probabilities to generate edges: train set with 60% nodes of size,val set with 20% nodes of size and test set with 20% nodes of size. We generate two different type graph as erdos renyi graph and random regular graph. The number of layer in experiment 1 and 2 in Table F and F is 2, for experiment 3 is 1.

**erdos renyi graph**: Erdos renyi graphs are random graphs with the number N of nodes and the probabilities p the generate edges while N and p have been set up beforehand. In this experiment we roughly maintain the number of edges in different graphs to be equal.

| | | random regular | | | erdos renyi | | |
|---|---|---|---|---|---|---|---|
| Method | aggregator | 4000 | 5000 | 6000 | 4000 | 5000 | 6000 |
| *GIN* | *mean* | $51.88 \pm 2.25$ | $55.2 \pm 1.5$ | $70.33 \pm 3.25$ | $58.37 \pm 6.25$ | $59.2 \pm 0.4$ | $75.42 \pm 0.5$ |
| | *max* | $51.88 \pm 2.25$ | $55.2 \pm 1.5$ | $70.33 \pm 3.25$ | $58.37 \pm 6.25$ | $59.2 \pm 0.4$ | $75.42 \pm 0.5$ |
| | *sum* | $56 \pm 6.37$ | $58.2 \pm 4.5$ | $70.33 \pm 3.25$ | $58.37 \pm 6.25$ | $59.2 \pm 0.4$ | $75.42 \pm 0.5$ |
| $2 - div\,GIN$ | *mean* | $71.37 \pm 0.87$ | $70.9 \pm 0.8$ | $68.17 \pm 3.17$ | $65.75 \pm 3$ | $65.7 \pm 2.6$ | $73.92 \pm 0.33$ |
| | *max* | $71.37 \pm 0.87$ | $70.9 \pm 0.8$ | $68.17 \pm 3.17$ | $65.75 \pm 3$ | $65.7 \pm 2.6$ | $73.92 \pm 0.33$ |
| | *sum* | **100** | **100** | **100** | **100** | **100** | **100** |

Table 3: Accuracy(%) of detecting triangle with different aggregators

| | | random regular | | | erdos renyi | | |
|---|---|---|---|---|---|---|---|
| layer | aggregator | 4000 | 5000 | 6000 | 4000 | 5000 | 6000 |
| 1 | *mean* | $71.37 \pm 0.87$ | $70.9 \pm 0.8$ | $68.17 \pm 3.17$ | $65.75 \pm 3$ | $65.7 \pm 2.6$ | $73.92 \pm 0.33$ |
| | *max* | $71.37 \pm 0.87$ | $70.9 \pm 0.8$ | $68.17 \pm 3.17$ | $65.75 \pm 3$ | $65.7 \pm 2.6$ | $73.92 \pm 0.33$ |
| | *sum* | **100** | **100** | **100** | **100** | **100** | **100** |
| 2 | *mean* | $71.5 \pm 0.25$ | $69.5 \pm 1.6$ | $66.5 \pm 3.25$ | $69.25 \pm 1.38$ | $64.8 \pm 2.4$ | $73 \pm 2.92$ |
| | *max* | $71.5 \pm 0.25$ | $69.5 \pm 1.6$ | $66.5 \pm 3.25$ | $69.25 \pm 1.38$ | $64.8 \pm 2.4$ | $73 \pm 2.92$ |
| | *sum* | **100** | **100** | **100** | **100** | **100** | **100** |
| 3 | *mean* | $70.62 \pm 0.5$ | $68.3 \pm 0.7$ | $64.2 \pm 3.16$ | $70.1 \pm 1.25$ | $66.2 \pm 0.32$ | $73 \pm 1.72$ |
| | *max* | $70.62 \pm 0.5$ | $68.3 \pm 0.7$ | $64.2 \pm 3.16$ | $70.1 \pm 1.25$ | $66.2 \pm 0.32$ | $73 \pm 1.72$ |
| | *sum* | **100** | **100** | **100** | **100** | **100** | **100** |

Table 4: Accuracy(%) of detecting triangle with different layers

| Method | aggregator | random regular | | | erdos renyi | | |
|---|---|---|---|---|---|---|---|
| | | 4000 | 5000 | 6000 | 4000 | 5000 | 6000 |
| $GIN$ | $mean$ | $64.58 \pm 0.42$ | $74.4 \pm 4.5$ | $79 \pm 1.5$ | $62.33 \pm 0.84$ | $75.4 \pm 1.3$ | $77.62 \pm 1.51$ |
| | $max$ | $64.58 \pm 0.42$ | $74.4 \pm 4.5$ | $79 \pm 1.5$ | $62.33 \pm 0.84$ | $75.4 \pm 1.3$ | $77.62 \pm 1.51$ |
| | $sum$ | $64.58 \pm 0.42$ | $74.4 \pm 4.5$ | $79 \pm 1.5$ | $62.33 \pm 0.84$ | $75.4 \pm 1.3$ | $77.62 \pm 1.51$ |
| $2 - div\,GIN$ | $mean$ | $71.5 \pm 0.25$ | $69.5 \pm 1.6$ | $66.5 \pm 3.25$ | $69.25 \pm 1.38$ | $64.8 \pm 2.4$ | $73 \pm 2.92$ |
| | $max$ | $71.5 \pm 0.25$ | $69.5 \pm 1.6$ | $66.5 \pm 3.25$ | $69.25 \pm 1.38$ | $64.8 \pm 2.4$ | $73 \pm 2.92$ |
| | $sum$ | **100** | **100** | **100** | **100** | **100** | **100** |

Table 5: Accuracy(%) of detecting triangle with different aggregators

**random regular graph**: A regular graph is a graph where each node has the same number of neighbors. In this experiment we roughly use $N \times p$ to set up the degree.

In undirected graph, if node v is contained by a triangle, then there exists a walk $(v, v_1, v_2, v)$. Notice $v_1$ is $v's$ $2_{(1)}$ neighbor, therefore a node is contained by a triangle if and only if there exists $v's$ $2_{(1)}$ neighbor. So task 1 can be expressed by 2-div graded modal logic as $\alpha_1$:

$$\alpha_1(x) = \Diamond^{\geq 1} y \tag{66}$$

Also we can deduce the formula node v is contained by at least two triangles can be expressed as:

$$\alpha_1(x) = \Diamond^{\geq 3} y \tag{67}$$

Hence Task 3 can be expressed by 2-div graded modal logic. There are also supplement experiment for experiment 3 with 2 layers.

