# OpenReview forum: "ENHANCEMENT OF GNN’S EXPRESSIVE POWER VIA RECONSIDERING MODAL LOGIC"
_ICLR.cc/2024/Conference — ICLR 2024 Conference Withdrawn Submission_

### Official Review · Reviewer_3sgu · 2023-10-13

**Soundness:** 1 poor
**Presentation:** 1 poor
**Contribution:** 2 fair
**Rating:** 3
**Confidence:** 5

**Summary:**

The paper studies the uniform expressive power of GNNs extended with information about k-hop neighborhoods. In particular, it generalizes results of Barceló et al. connecting the uniform expressive power of GNNs and graded modal logic.

**Strengths:**

The paper deals with an interesting problem.

Some of the results seem non-trivial: there is potential on the paper.

**Weaknesses:**

Unfortunately, the presentation of this paper is horrible. Not only many sentences are broken, but also the paper is full of typos and undefined notions. The english is extremely poor, which makes the understanding of the notions introduced simply impossible. I declare myself to be an expert on this topic, but this paper I could not understand at all. As an example, one of the main theorems of the paper, the one claiming that a certain logic coincides with GNNs, is presented without having introduced GNNs before!

I have the feeling that there is something interesting, and even deep, in this paper, but the authors need to work extensively on the presentation for this to become clear. As such, it is not at the level of what I'd consider "reviewable" for NeurIPS.

**Questions:**

No questions. I suggest the paper to be rewritten extensively.

---

### Official Review · Reviewer_KZTv · 2023-10-31

**Soundness:** 2 fair
**Presentation:** 1 poor
**Contribution:** 1 poor
**Rating:** 1
**Confidence:** 4

**Summary:**

The paper proposes an enhanced framework for studying message passing neural networks, and an enhanced graded modal logic to recreate results of Barceló et al 2020.
I found the paper quite difficult to read, most of it due to shortcomings of the authors in making the definitions clear; see for example the definition of k-hop (l_1,...,) neighbourhood. But if I understood correctly, the proposal is to create masks of the local structure of each node, but where masks are given in terms of adding, or removing, certain nodes from the graph. If so, then the paper has two main weak points. First, resulting GNNs are not permutation-invariant (unless one permutes everything in the GNNs accordingly, but this is not treated in the paper). And second, the paper misses comparation on more recent literature regarding similar techniques.

**Strengths:**

* New model of GNNs based on masking certain nodes in the graph.

**Weaknesses:**

* Proposal assumes a given adjacency matrix and therefore resulting GNN architectures are not permutation-invariant. This is extremely important, and If one is willing to loose permutation invariance then probably a transformer is the way to go for graphs.
* The proposal should probably fit in the framework given by Qian et al  in Neurips 2022 (altough the latter retains permutation invariance).
* Paper is difficult to read, and definitions / notation is not clear enough. For example, how do I distinguish from the neighbourhood of a node when I mask just the node v_1, and the neighbourhood when I mask just the node v_2?

**Questions:**

GNN models that are not permutation invariant should be much better motivated, I think a good comment in this direction is missing.

---

### Official Review · Reviewer_oNW8 · 2023-11-01

**Soundness:** 2 fair
**Presentation:** 1 poor
**Contribution:** 2 fair
**Rating:** 3
**Confidence:** 2

**Summary:**

In this paper the connection between GNNs and modal logic, as outlined in Barceló et al, is extended using a more complicated higher-order modality. In particular, instead of the simple edge modality, exponentially many modalities are considered, each of them encoding directed path (non)-existences between nodes. It is shown that logical classifiers expressible by an extended notion of GNNs correspond precisely to those expressible in the extended modal logic, hereby generalising the result by Barceló et al. A consequence is that the extended modalities and GNNs have higher expressive power.

**Strengths:**

1. The use of extended modalities and their use to construct expressive GNN is novel.

2. The main result requires some non-trivial analysis of logical formulae.

**Weaknesses:**

1. The paper is not well written and very difficult to follow. The authors should present their work in a more clear way.

2. Large part of the paper (including all experiments and all comparisons with recent higher order GNNs) are deferred to the appendix. I did not consult the appendix.

3. The proposed approach depends on the introduction of exponentially many different neighborhood (exponential in l). This limits the practically relevance for higher values of ell.

**Questions:**

**Q** Please explain more clearly how you can check triangles using your method. Section 3.2 is the main part of the paper but it is very badly presented.